# Disruption of Cav1.2-mediated signaling is a pathway for ketamine-induced pathology

Huan Chen[1], David H. Vandorpe[1], Xiang Xie[1], Seth L. Alper [1], Mark L. Zeidel[1] & Weiqun Yu[1]✉

The general anesthetic ketamine has been repurposed by physicians as an anti-depressant and by the public as a recreational drug. However, ketamine use can cause extensive pathological changes, including ketamine cystitis. The mechanisms of ketamine's anti-depressant and adverse effects remain poorly understood. Here we present evidence that ketamine is an effective L-type $Ca^{2+}$ channel (Cav1.2) antagonist that directly inhibits calcium influx and smooth muscle contractility, leading to voiding dysfunction. Ketamine prevents Cav1.2-mediated induction of immediate early genes and transcription factors, and inactivation of *Cav1.2* in smooth muscle mimics the ketamine cystitis phenotype. Our results demonstrate that ketamine inhibition of Cav1.2 signaling is an important pathway mediating ketamine cystitis. In contrast, Cav1.2 agonist Bay k8644 abrogates ketamine-induced smooth muscle dysfunction. Indeed, Cav1.2 activation by Bay k8644 decreases voiding frequency while increasing void volume, indicating Cav1.2 agonists might be effective drugs for treatment of bladder dysfunction.

[1] Department of Medicine, Beth Israel Deaconess Medical Center, Harvard Medical School, Boston, MA, USA. ✉email: wyu2@bidmc.harvard.edu

The anesthetic ketamine has been increasingly used for treatment of depression and pain, and increasingly subject to recreational use and abuse for its mood-elevating effects over the last 20 years[1,2]. Ketamine is well established as a N-methyl-D-asparate receptor (NMDAR) antagonist, buts its anti-depressant and mood-elevating effects remain poorly understood. Ketamine can inhibit NMDAR-dependent bursting activity of lateral habenula neurons to elevate mood[3], while the ketamine metabolite, hydroxynorketamine, activates the α-amino-3-hydroxy-5-methyl-4-isoxazolepropionic acid receptor (AMPAR)[4]. Although both NMDAR and AMPAR are highly expressed in neurons, neither NMDAR inhibition nor AMPAR activation likely explain the broad spectrum of acute and chronic adverse effects of ketamine, including cognitive and memory impairment, hypertension, renal and urological dysfunction, hepatobiliary abnormalities, gastrointestinal symptoms, and many other recently recognized disorders[5–8]. Thus, ketamine's adverse effects might reflect action on one or more previously unrecognized receptors.

Ketamine cystitis is one of the most common among severe ketamine-induced pathologies, affecting up to 30% of long-term users[5,8–10]. Since first reported in 2007[11], thousands of subsequently reported cases have left uncounted many more unreported and/or unrecognized ketamine abusers throughout the world[5,8,10]. The major manifestation of ketamine cystitis is intolerable bowel or bladder pain, accompanied by urinary urgency, frequency, nocturia, dysuria, and even hematuria. Urodynamic and ultrasonic studies reveal a contracted bladder with detrusor instability and decreased bladder capacity (as low as 10–20 ml per void), resulting in severe urge incontinence with napkin-dependence. Fifty percent of ketamine cystitis patients have hydronephrosis with impaired kidney function.

The molecular mechanism of ketamine-induced pathological changes remains unclear. Cystoscopy reveals inflammation with lymphocyte infiltration, mucosal ulceration, urothelial atypia and thinning. Pathophysiological studies have shown increased inflammation, dedifferentiation, and apoptosis in patient bladder, in human cell lines, and in animal models[12–16]. Proposed pathological mechanisms have included direct damage to urothelial barrier function, increased autoimmune response, ischemic microvascular toxicity, smooth muscle myocyte vacuolation, and neuronal degeneration or hyperplasia[17]. Although urothelium is in direct contact with high concentrations of ketamine in the urine, and disruption of the urothelial barrier has been reported in both patients and animal models, we recently observed no effect on urothelial structure and permeability from 12 weeks of ketamine therapy that nonetheless reduced bladder compliance, void volume, and voiding intervals[18]. Similar findings in human patients and animal models suggest that disrupted urothelial barrier function may not be the direct cause of ketamine cystitis[14,18,19].

The few current approaches for treatment of ketamine cystitis reflect the lack of a generally accepted mechanism. Abstinence from ketamine use is the most effective treatment, whereas anti-inflammatory drugs, anticholinergics, pentosan polysulfate, intravesical glycosaminoglycan preparations, and other supportive therapies have been of limited benefit. This lack of effective medical therapy can lead to cystectomy and bladder reconstruction. Thus, mechanistic understanding of ketamine cystitis and other adverse effects of ketamine constitutes an urgent unmet need, and is a prerequisite for development of effective therapies.

Here, we present evidence that ketamine causes functional and pathological changes in smooth muscle by inhibiting the L-type voltage-gated calcium channel Cav1.2, and that activation of this channel can reverse these pathological changes.

## Results

**Ketamine inhibits BSM contractility.** Ketamine and its metabolite norketamine are both excreted by the kidney, with reports of urinary ketamine/norketamine concentrations up to ~50 μg ml$^{-1}$ after therapeutic use or recreational abuse[20–23]. Although ketamine cystitis has been attributed to urinary ketamine toxicity, this hypothesis has never been tested by direct ketamine instillation into the bladder. We therefore performed cystometrogram (CMG) studies during ketamine infusion into mouse bladder. Ketamine instillation induced acute voiding dysfunction phenotypes typical of ketamine cystitis, including decreased voiding pressure, decreased voiding interval, decreased bladder compliance, and sometimes bladder spasm (Supplementary Fig. 1). These smooth muscle-related abnormalities suggested direct inhibition of ketamine on bladder smooth muscle (BSM) function. Indeed, our myographic studies detected inhibition of BSM contraction at sub-μg ml$^{-1}$ concentrations of ketamine, with complete inhibition between 100 and 500 μg ml$^{-1}$ (Fig. 1a, b). These data are consistent with plasma ketamine concentrations in both ketamine abusers and patients, with plasma concentrations of 0.1 μg ml$^{-1}$ producing analgesia, 0.05–0.2 μg ml$^{-1}$ drowsiness and perceptual distortions, and 2–3 μg ml$^{-1}$ for general anesthesia, with reported anesthetic concentrations as high as 26 μg ml$^{-1}$ [24,25]. In chronic ketamine abusers, ketamine consumption up to 28 g per day has been reported, with urine ketamine/norketamine concentrations up to 25/50 μg ml$^{-1}$ [23,26]. BSM contraction was also inhibited by the major ketamine metabolite, norketamine, but not by hydroxynorketamine (Supplementary Fig. 2). Norketamine exhibits lower potency than ketamine as an NMDAR antagonist.

To rule out ketamine action via inhibition of neurotransmitter release from local nerve fibers, ketamine was tested in the presence of direct BSM contractile agonists carbachol (an M2/M3 cholinergic receptor ligand) and α,β-meATP (a P2X$_1$ purinergic receptor ligand). BSM contraction elicited by these direct agonists was inhibited by ketamine in a dose-dependent manner (Fig. 1c–h), consistent with ketamine's direct inhibition of BSM contraction in response to electrical field stimulation (EFS).

In addition to severe urinary tract symptoms, ketamine induces dysfunction of gastrointestinal and hepatobiliary tracts, and the vasculature as each is surrounded by smooth muscle. We therefore performed myographic studies on gastric and jejunal smooth muscle, and found inhibition by ketamine of enteric smooth muscle contraction in a dose-dependent manner (Supplementary Figs. 3, 4), suggesting smooth muscle as an important target for ketamine-induced pathology.

**Ketamine inhibition of BSM contraction is NMDAR-independent.** Ketamine is a noncompetitive NMDAR antagonist, with a sub-μg ml$^{-1}$ IC$_{50}$. NMDAR subunit mRNA and protein are expressed in male lower urogenital tract[27], but expression has not been reported in BSM. We therefore characterized the pharmacology of BSM strip contraction. The NMDAR agonists (NMDA and RS-Tetrazol-5-yl glycine) at concentrations up to 100 μM had no effect on EFS-stimulated BSM contractile force (Supplementary Fig. 5a, b). Moreover, NMDAR antagonists D-AP5, CGS 19755, and MK 801 had little or no effect on BSM contraction (Supplementary Fig. 5c–e).

To further test a possible role of NMDAR in BSM contraction, we created a smooth muscle-specific knockout mouse strain for the functionally essential NMDAR subunit, NR1 (*SMNR1KO*) by mating *NR1$^{f/f}$* mice (*B6.129S4-Grin1$^{tm2Stl}$/J*) with *Sm22α-cre* mice (*B6.Cg-Tg(Tagln-cre)1Her/J*). These smooth muscle-specific NR1 knockout mice were grossly normal, with wild type voiding function as indicated by voiding spot assay (VSA) (Fig. 2a–c). Furthermore, these mice exhibit normal BSM contraction force in

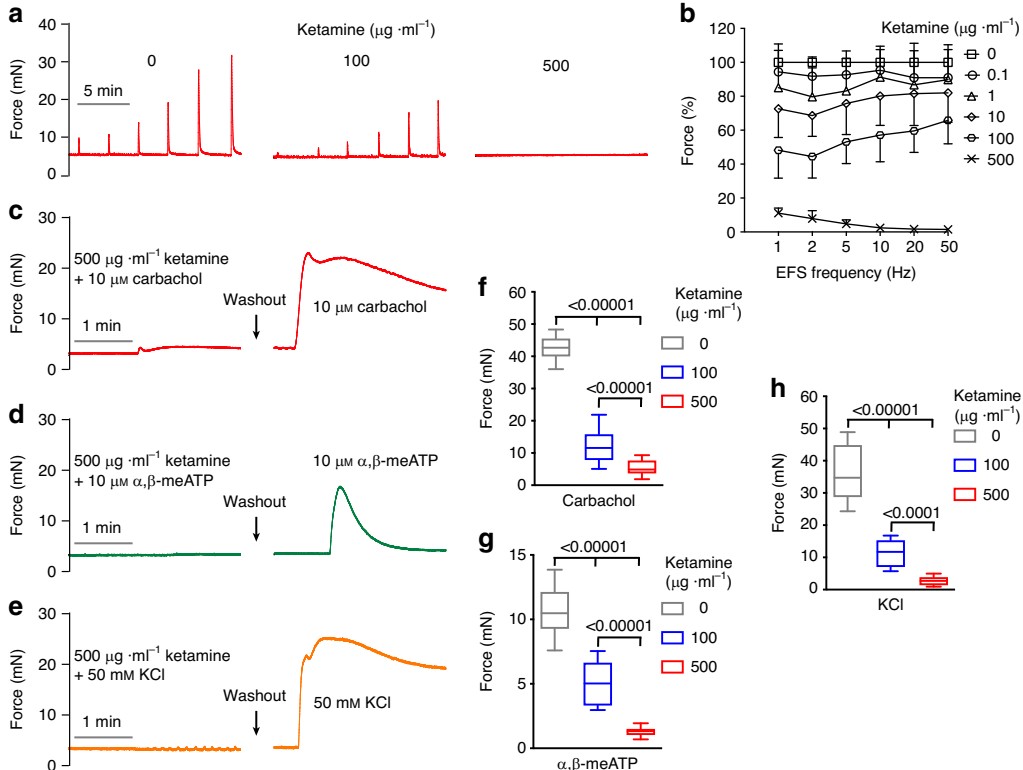

**Fig. 1 Ketamine inhibits BSM contraction. a** Representative traces of mouse BSM contraction in response to EFS (1, 2, 5, 10, 20, 50 Hz), showing dose-dependent inhibition by ketamine (0.1–500 μg ml$^{-1}$). **b** Summarized data from experiments as in a ($n = 12$ BSM strips). Data are presented as mean values ± SD. **c–e** Representative traces of mouse BSM treated first with ketamine plus either carbachol (**c** $n = 14$ BSM strips), α,β-meATP (**d** $n = 10$ BSM strips), or 50 mM KCl (**e** $n = 11$ BSM strips), followed by drug washout and subsequent re-exposure to the indicated drugs in the absence of ketamine. **f–h** Summarized data from **c–e**. Data are shown as box and whiskers, center line is the median of the data set, box represents 75% of the data, and bars indicates whiskers from minimum to maximum. Data were analysed by one-way ANOVA with Bonferroni's posthoc tests. $P < 0.05$ is considered to be significant and $P$ values are given above the bars. Source data are provided as a Source Data file.

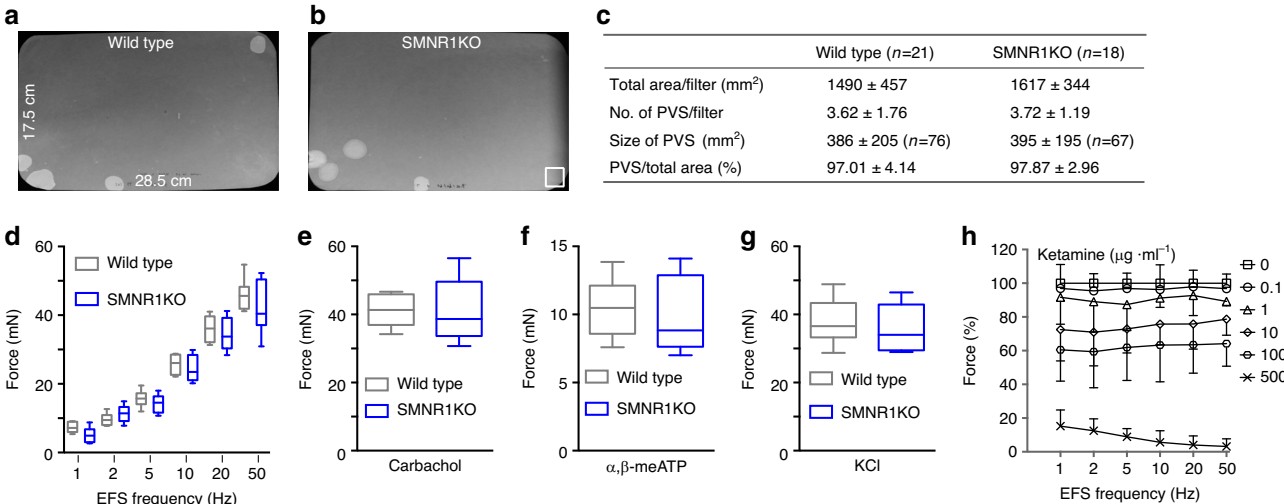

**Fig. 2 Ketamine inhibition of BSM contraction is not mediated by NMDAR.** Representative filters show UV light-illuminated urine spots from a wild type mouse (**a** $n = 21$ filters) and a smooth muscle-specific NR1 knockout (SMNR1KO) mouse (**b** $n = 18$ filters). Summarized quantitative data (**c**) indicate normal voiding volume, normal spot number per void, and normal spot size for SMNR1KO mice. ± in **c** is SD of the mean. In **b** 400 mm$^2$ white box at bottom right serves as area standard. **d–g** BSM contraction in SMNR1KO mice stimulated by EFS (**d** wild type $n = 8$ BSM strips, SMNR1KO $n = 8$ BSM strips), by 10 μM carbachol (**e** wild type $n = 8$ BSM strips, SMNR1KO $n = 8$ BSM strips), by 10 μM α,β-meATP (**f** wild type $n = 8$ BSM strips, SMNR1KO $n = 8$ BSM strips), or by 50 mM KCl (**g** wild type $n = 8$ BSM strips, SMNR1KO $n = 8$ BSM strips). Data are shown as box and whiskers, center line is the median of the data set, box represents 75% of the data, and bars indicates whiskers from minimum to maximum. Data were analysed by Student's $t$-test. **h** ketamine concentration-dependent inhibition of contraction of BSM from SMNR1KO mice ($n = 12$ BSM strips). The data indicate that NMDAR does not mediate ketamine-induced inhibition of BSM contraction. Data are presented as mean values ± SD. Source data are provided as a Source Data file.

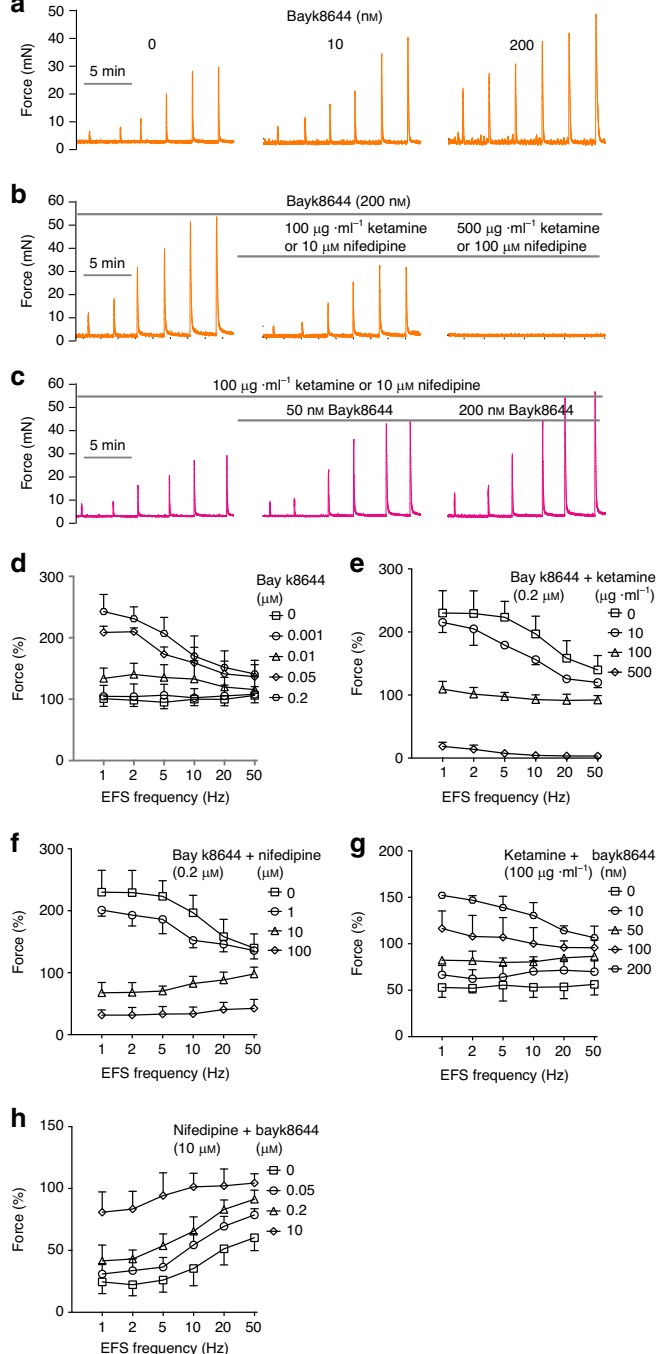

**Fig. 3 Ketamine and Cav1.2 agonist Bay k8644 are mutual antagonists. a** Representative traces of Bay k8644-mediated potentiation of EFS-stimulated BSM contraction, summarized in **d** ($n = 7$ BSM strips). **b** Representative traces of Bay k8644-mediated potentiation of EFS-stimulated BSM contraction fully inhibited by ketamine and by nifedipine in dose-dependent manners, summarized in **e** ($n = 8$ BSM strips) and **f** ($n = 12$ BSM strips). **c** representative traces of ketamine-induced and nifedipine-induced inhibition of EFS-stimulated BSM contraction rescued by 50 or 200 nM Bay k8644, summarized in **g** ($n = 17$ BSM strips) and **h** ($n = 13$ BSM strips). Data are presented as mean values ± SD. These data support ketamine as an inhibitor of Cav1.2-mediated BSM contraction. Source data are provided as a Source Data file.

response to EFS, carbachol, α,β-meATP, and KCl (Fig. 2d–g). EFS-elicited contractile force in the *SMNR1KO* mouse retained dose-dependent inhibition by ketamine (Fig. 2h). These data suggest that NMDAR plays no role in BSM contractility, and thus is not likely a major mediator of ketamine cystitis.

**Ketamine inhibits Cav1.2-mediated calcium channel activity.** Ketamine's dose-dependent, NMDAR-independent inhibition of BSM contraction in response to EFS and to muscarinic/purinergic agonists (Figs. 1 and 2) suggests ketamine blockade of a critical contractile pathway target downstream of both muscarinic and purinergic signaling. The L-type calcium channel Cav1.2 was shown to be essential for urinary BSM contractility, and mouse BSM deficient in Cav1.2 lacked contractile responses to stimulation by carbachol and by high K+[28,29]. As with ketamine inhibition of BSM contraction, the Cav1.2-specific antagonist, nifedipine, inhibited BSM contraction induced by EFS and by muscarinic and purinergic agonists (Supplementary Fig. 6). We therefore hypothesized that ketamine might act as a Cav1.2 antagonist to inhibit BSM contraction, leading to pathological consequences. Indeed, both ketamine and nifedipine inhibited BSM contraction elicited by depolarization with 50–100 mM extracellular KCl, to a degree adequate to activate voltage-dependent Cav1.2. (Supplementary Fig. 6). Furthermore, Cav1.2 agonist Bay k8644-potentiated BSM contraction was dose-dependently inhibited by ketamine as well as by nifedipine. Moreover, prior inhibition of BSM contraction by nifedipine or by ketamine was dose-dependently reversed by subsequent exposure to Cav1.2 agonist Bay k8644 (Fig. 3). These data indicate mutual antagonism of ketamine and Bay k8644, suggesting ketamine inhibits BSM contraction through its action as a Cav1.2 antagonist.

To further explore the hypothesis that ketamine directly inhibits Cav1.2, we tested ketamine's inhibition of Cav1.2 expressed in *Xenopus* oocytes (from cRNAs encoding Cav1.2 subunits mouse $\alpha_1$, rat $\alpha_2\delta_1$, and rat $\beta_3$). Seventy-two hours post-cRNA injection, oocytes were subjected to two-microelectrode voltage clamp in a $Ca^{2+}$-free bath containing 10 mM $Ba^{2+}$ solution[30]. As shown in Fig. 4a–e, whereas water-injected oocytes exhibited no apparent inward $Ba^{2+}$ current, oocytes previously injected with Cav1.2 subunit cRNAs displayed significant inward $Ba^{2+}$ current, indicating robust functional expression of Cav1.2 sensitive to inhibition by both ketamine and nifedipine (Fig. 4d, e). To further test ketamine inhibition of Cav1.2, freshly isolated mouse BSM cells were subjected to whole cell patch clamp recording. Ketamine consistently inhibited Cav1.2-mediated $Ba^{2+}$ currents (Fig. 4f–h). Inward currents induced in freshly isolated BSM cells by exposure to 100 nM Cav1.2 agonist Bayk8644 were completely inhibited by 500 μg ml⁻¹ ketamine added in the continued presence of Bayk8644 (Suppl. Fig. 7a, d), consistent with ketamine effects on *Xenopus* oocytes (Fig. 4), on BSM strip contraction force (Fig. 1), and on Bayk8644-induced muscle contraction (Fig. 3e). Ketamine inhibition of BSM cell Cav1.2 activity is mediated in part by inducing a left-shift in voltage-dependent inactivation (Supplementary Fig. 7h) without significant change in time constant of inactivation (Supplementary Fig. 7e, h).

Furthermore, in primary cultured mouse BSM (mBSM) cells, Bay k8644-induced $Ca^{2+}$ influx and intracellular $Ca^{2+}$ signaling were similarly blocked by both ketamine and by nifedipine (Fig. 5). Ketamine and nifedipine also inhibited $Ca^{2+}$ influx and intracellular $Ca^{2+}$ signaling induced by treatment of mBSM cells with KCl, carbachol, or ATP (Fig. 5). Ketamine and nifedipine also inhibited $Ca^{2+}$ influx and intracellular $Ca^{2+}$ signaling elicited by Bay k8644, by KCl depolarization, by carbachol, or

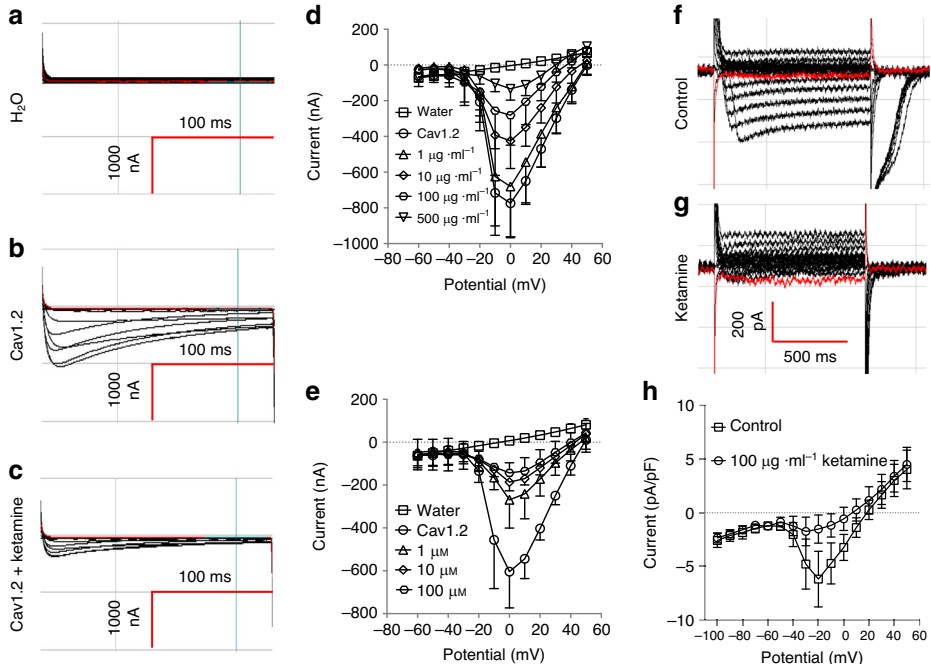

**Fig. 4 Ketamine inhibits Cav1.2-mediated calcium channel activity. a–c** are representative $Ba^{2+}$ current traces of two microelectrode voltage-clamped *Xenopus* oocytes, previously injected with water (**a**, $n = 5$ oocytes) or with cRNA encoding Cav1.2 subunits, recorded in the absence (**b**, $n = 10$ oocytes), or presence of ketamine (**c** $n = 10$ oocytes). **d, e** summarize data from **a–c** showing dose-dependent inhibition of Cav1.2-mediated $Ba^{2+}$ current by ketamine (**d** $n = 10$ oocytes) or nifedipine (**e** $n = 5$ oocytes). **f, g** are representative whole cell current traces of fresh isolated BSM cells in the absence (**f** $n = 8$ BSM cells) or presence (**g** $n = 8$ BSM cells) of 100 μg ml$^{-1}$ ketamine. **h** Summarizes data showing ketamine inhibition on Cav1.2-mediated $Ba^{2+}$ current in freshly isolated mouse BSM cells as shown in **f, g**. Data are presented as mean values ± SD. Source data are provided as a Source Data file.

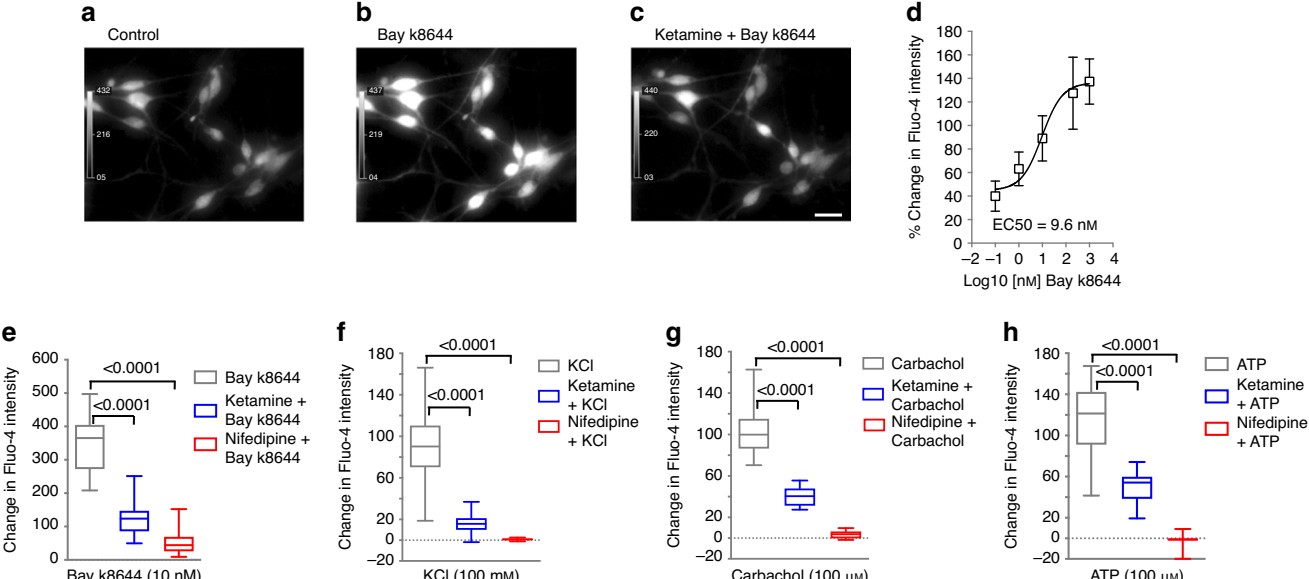

**Fig. 5 Ketamine inhibits Cav1.2-mediated calcium influx in mouse BSM cells. a–c** Representative Fluo-4 $Ca^{2+}$ images of primary cultured mouse BSM cells treated without (**a**) or with Bay k8644 (10 nM) (**b**), or treated first with ketamine (100 μg ml$^{-1}$) and then with added Bay k8644 (10 nM) (**c**). Scale bar: 40 μm. **d** Nonlinear curve fit of Bay k8644 concentration-dependent increase in mouse BSM cell $[Ca^{2+}]_i$ with EC$_{50}$ of 9.6 nM ($n = 10$, 23, 18, 19, 10, and 29 BSM cells for respective [Bay k8644] = 0, 0.1, 1, 10, 200, and 1000 nM). Bars indicate the SD of the means. **e** 10 nM Bay k8644-induced increase in BSM intracellular $[Ca^{2+}]$ ($n = 39$ BSM cells) was inhibited by 100 μg ml$^{-1}$ ketamine ($n = 25$ BSM cells)) and by 10 μM nifedipine ($n = 39$ BSM cells). **f** Hundred millimolar of KCl-stimulated increase in BSM intracellular $[Ca^{2+}]$ ($n = 39$) was inhibited by 100 μg ml$^{-1}$ ketamine ($n = 25$ BSM cells) and by 10 μM nifedipine ($n = 39$ BSM cells). **g** Hundred micromolar of carbachol-stimulated increase in BSM intracellular $[Ca^{2+}]$ ($n = 27$ BSM cells) was inhibited by 100 μg ml$^{-1}$ ketamine ($n = 27$ BSM cells), and by 10 μM nifedipine ($n = 30$ BSM cells). **h** Hundred micromolar of ATP-stimulated increase in BSM intracellular $[Ca^{2+}]$ ($n = 43$ BSM cells) was inhibited by 100 μg ml$^{-1}$ ketamine ($n = 22$ BSM cells) and by 10 μM nifedipine ($n = 21$ BSM cells). Data are shown as box and whiskers, center line is the median of the data set, box represents 75% of the data, and bars indicates whiskers from minimum to maximum. Data were analysed by two tailed Student's *t*-test. $P < 0.05$ is considered to be significant and *P* values are given above the bars. Source data are provided as a Source Data file.

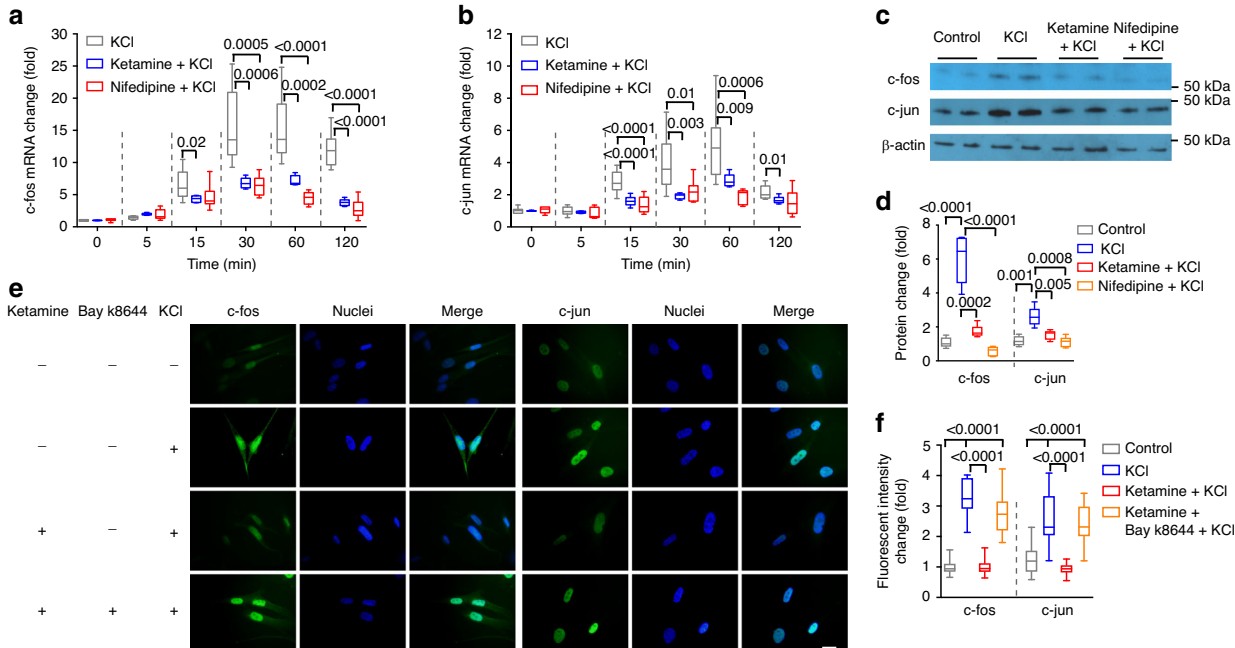

**Fig. 6 Ketamine reduces Cav1.2-stimulated mRNA and protein levels in BSM cells.** Mouse BSM strips were subjected to 50 mM KCl activation of Cav1.2 for 0–120 min and then lysed for mRNA preparation. Changes in *c-fos* (**a** *n* = 9 BSM strips) and *c-jun* (**b** *n* = 9 BSM strips) mRNA were measured by quantitative RT-PCR. Upregulation of both *c-fos* and *c-jun* were inhibited by 100 µg ml⁻¹ ketamine and by 10 µM nifedipine. **c** Representative immunoblot from two independent experiments showing that upregulation of c-fos and c-jun by 30 min treatment with 50 mM KCl to activate Cav1.2 was inhibited by ketamine (100 µg ml⁻¹) and by nifedipine (10 µg ml⁻¹). **d** summarized data (*n* = 5 BSM strips) of experiments similar to that shown in **c**. **e** 50 mM KCl upregulated expression and nuclear translocation of c-fos and c-jun in human BSM cells in a manner inhibited by ketamine (100 µg ml⁻¹) and rescued by Bay k8644 (200 nM). Scale bar: 20 µm. These images, representative of two separate triplicate experiments, are summarized in **f** (*n* = 38–46 BSM cells). Data are shown as box and whiskers, center line is the median of the data set, box represents 75% of the data, and bars indicates whiskers from minimum to maximum. Data are analysed by two tailed Student's *t*-test. *P* < 0.05 is considered to be significant and *P* values are given above the bars. Source data are provided as a Source Data file.

by ATP in primary cultured human BSM (hBSM) cells (Supplementary Fig. 8). These data conclusively identify ketamine as a Cav1.2 antagonist which can block BSM cell Ca²⁺ influx and inhibit BSM contractility.

**Ketamine disrupts Cav1.2-regulated transcription.** Cav1.2 channels play central roles in regulating biochemical and electrical signaling of neurons and muscle cells[31]. Ca²⁺ influx through Cav1.2 activates neuronal Ca²⁺-dependent signaling proteins which transmit signals into the nucleus to regulate transcription factors such as CREB, MEF, and NFAT. Ca²⁺ entering through Cav1.2 channels can also diffuse to the nucleus to activate nuclear calcium-dependent enzymes such as CaMKIV, which in turn regulate activities of transcription factors and transcriptional coregulators. Neuronal *c-fos* and *c-jun* are well documented immediate early genes (IEGs) upregulated in response to Cav1.2 activation[32–36]. *c-fos* and *c-jun* in urinary bladder are up-regulated in response to mechanical stretch, suggesting a role of these IEGs in maintaining normal bladder function during cyclic filling and emptying[37]. We thus hypothesized that ketamine induced pathological changes by inhibition of Cav1.2-mediated transcription. Indeed, Cav1.2 activation in mouse BSM strips by 50 mM KCl depolarization increased BSM mRNA and protein levels of both *c-fos* and *c-jun*. These changes were blocked by 15 min pretreatment with either ketamine or nifedipine before KCl depolarization (Fig. 6a–d), and were replicated in primary cultured hBSM cells (Supplementary Fig. 9). Further immunostaining studies confirmed that ketamine and nifedipine inhibited nuclear translocation of c-fos and c-jun in hBSM cells, and that Bay k8644 rescued this inhibition (Fig. 6e, f).

KCl depolarization-mediated activation of BSM cell Cav1.2 also upregulated CREB, MEF, NFAT, and FoxO transcription factors in manners sensitive to inhibition by pretreatment with ketamine or nifedipine (Supplementary Figs. 11–14). Ketamine and nifedipine also inhibited stretch-induced upregulation of c-fos and c-jun in mouse BSM cells (Supplementary Fig. 10)[37], suggesting that mechanical stretch during bladder filling and voiding modulates bladder function in a Cav1.2-dependent manner.

IEGs and transcription factors modulated by Cav1.2 signaling are important for cell survival, plasticity, proliferation, and differentiation. Ketamine cystitis patients commonly present with smaller bladders and decreased body weight, findings reproduced in many ketamine cystitis animal model studies. We thus hypothesize that ketamine-inhibited Cav1.2 signaling decreases BSM proliferation, in part through inhibition of c-fos and c-jun action. To test this hypothesis, hBSM cells were incubated for up to 7 days in the presence of ketamine or nifedipine at a range of concentrations, then assessed for proliferation. Control cells proliferated until reaching plateau at day 4–5. In contrast, ketamine and nifedipine both inhibited hBSM cell proliferation, and at high concentrations led to cell death. (Fig. 7a, b). This inhibition of BSM proliferation was validated by the observation that ketamine and nifedipine each reduced Ki67-positive cell numbers in a manner reversed by Bay k8644 treatment (Fig. 7c), supporting the importance of Cav1.2 signaling in ketamine toxicity.

**Cav1.2 inactivation mimics the ketamine cystitis phenotype.** Cav1.2 is essential for BSM contractility, and mice BSM deficient in Cav1.2 lacked contractile responses to carbachol and to high

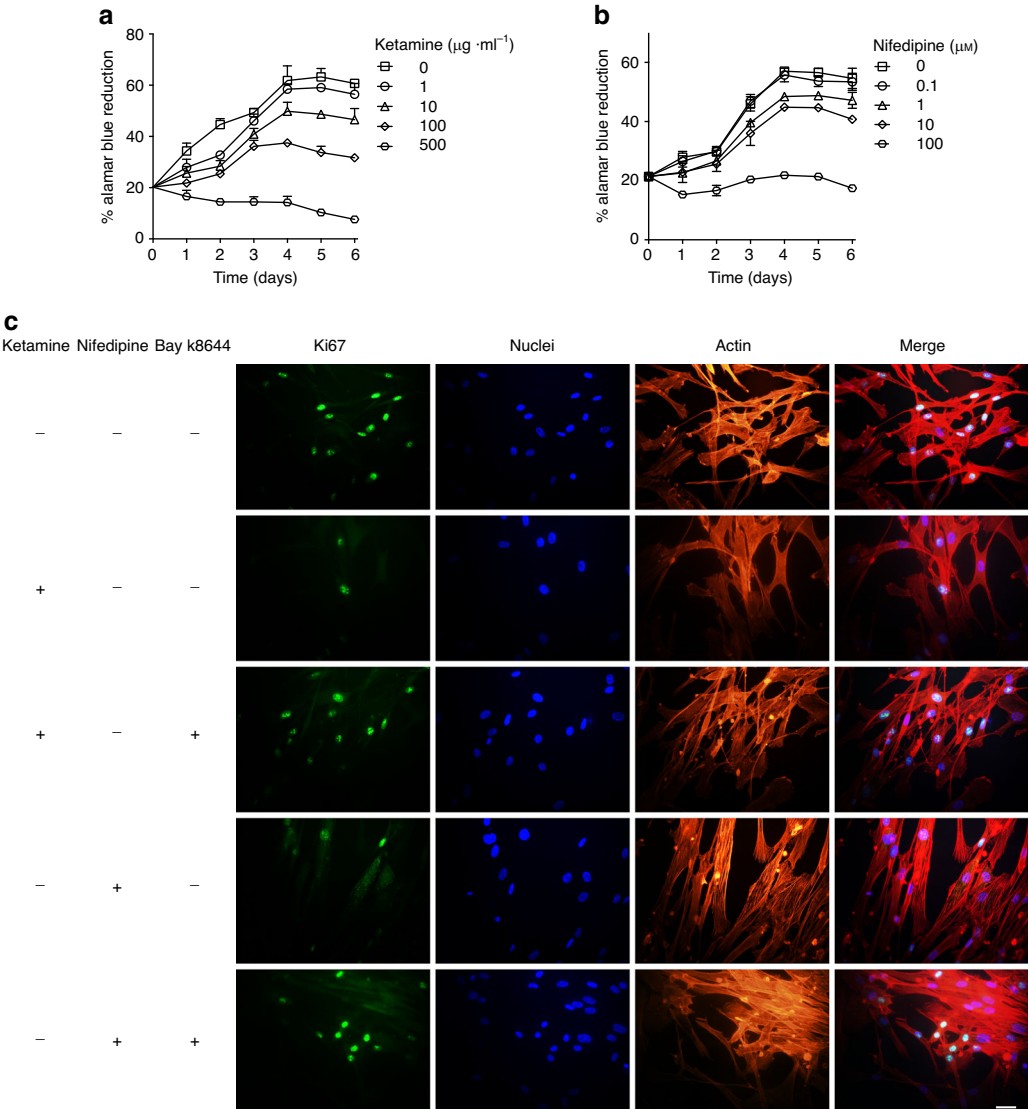

**Fig. 7 Ketamine and nifedipine dose-dependently inhibit human BSM cell proliferation.** Primary cultured human BSM cells were incubated with ketamine or nifedipine for 1–7 days at the indicated concentrations, cell proliferation was assessed by alamar blue assay in the presence of ketamine (**a** $n = 4$) or nifedipine (**b** $n = 4$). Ketamine and nifedipine inhibited hBSM cell proliferation. **c** Immunofluorescent staining of ki67 (green) in hBSM cells was inhibited by pretreatment with ketamine (100 μg ml$^{-1}$) or nifedipine (10 μM), but inhibition was rescued by 200 nM Bay k8644, representative images are from two independent experiments with at least three samples each group at each experiment. Scale bar: 40 μm. Data are presented as mean values ± SD. Source data are provided as a Source Data file.

K$^+$ depolarization[28,29]. In ketamine cystitis patients, urine (or plasma) ketamine concentrations likely inhibit Cav1.2 function partially. To mimic partial inhibition of Cav1.2, we generated *SMCav1.2*$^{+/-}$ mice with heterozygous knockout of Cav1.2 in smooth muscle by crossing mice carrying Cre recombinase under the control of the smooth muscle transgelin promoter (*SM22α-creKI*) with *Cav1.2*$^{fl/fl}$. *SMCav1.2*$^{+/-}$ BSM exhibited reduced Cav1.2 mRNA and protein (Supplementary Fig. 15), and responded to individual stimulation by EFS, carbachol, and α,β-meATP with decreased contractile force partially reversible by Bay k8644 (Fig. 8h–m). *SMCav1.2*$^{+/-}$ BSM cells in primary culture showed reduced [Ca$^{2+}$]$_i$ in response to stimulation by high K$^+$, carbachol, ATP, or Bay K8644 (Supplementary Fig. 16). The reduced size of *SMCav1.2*$^{+/-}$ mouse bladders, accompanied by reduced bladder wall thickness and smooth muscle cell size (Fig. 8a–g), resembled human patients and animal models with ketamine cystitis in bladder size and histopathology, and in decreased c-fos and c-jun mRNA and protein levels

(Supplementary Fig. 17). In vivo VSA and CMG studies in *SMCav1.2*$^{+/-}$ mice indicated bladder overactivity with reduced bladder volume, contractile pressure and compliance (Fig. 9). Thus, Cav1.2 haploinsufficiency in smooth muscle decreased BSM contractile force and growth, leading to an overactive bladder phenotype recapitulating ketamine cystitis in humans.

To rule out possible developmental compensations to the phenotype of the constitutively haploinsufficient *SMCav1.2*$^{+/-}$ mouse model, we created a tamoxifen-inducible smooth muscle-specific *Cav1.2* knockout (*TISMCav1.2*$^{-/-}$) mouse model by crossing smooth muscle myosin heavy chain promoter-Cre (*SMMHC-CreER*$^{T2}$, Jackson laboratory, Maine) males with *Cav1.2*$^{fl/fl}$ females. *Cav1.2* genes were selectively deleted in smooth muscle by intraperitoneal injection of 4-hydroxytamoxifen (OHT, 40 mg/kg/day) for five consecutive days. Mice were phenotyped two weeks after injection for *Cav1.2* mRNA (qPCR) and protein (immunoblot) to verify the BSM genotype (Supplementary Fig. 15d–f). *TISMCav1.2*$^{-/-}$ mice

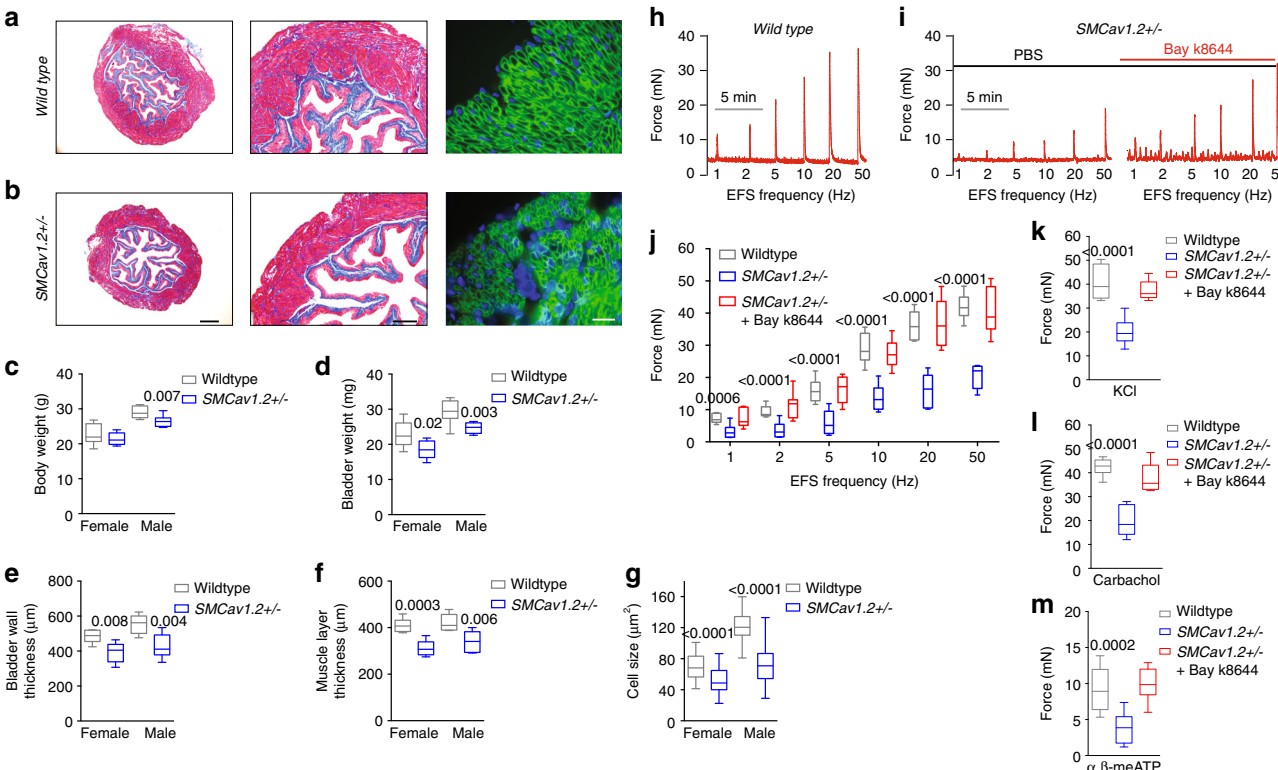

**Fig. 8 SMCav1.2+/− mice exhibit abnormal BSM morphology and contractility. a, b** Representative images from three independent experiments: Masson's trichrome stained bladder sections show that SMCav1.2$^{+/-}$ mouse bladders are smaller with a thinner BSM layer. Scale bars are, from left to right, 400 μm, 200 μm, and 20 μm, respectively. The summarized data are shown in **c–f. c, d** (n = 8 mice for both wildtype and SMCav1.2$^{+/-}$ groups) show that SMCav1.2$^{+/-}$ mice have lower body weight and bladder weight. **e, f** (n = 6 mice for both wildtype and SMCav1.2$^{+/-}$ groups) show that SMCav1.2$^{+/-}$ mice have reduced bladder wall and correspondingly lower muscle layer thickness. Individual BSM cells were subjected to immunofluorescence staining and imaged with antiβ1 integrin antibody (green) and DAPI for nuclei (blue). BSM cell cross sectional area was quantitated using Fiji software. SMCav1.2$^{+/-}$ mice BSM cell size was smaller (**g** n = 32, 54, 55, and 73 cells for wildtype female, SMCav1.2$^{+/-}$ female, wildtype male, and SMCav1.2$^{+/-}$ male mice groups, respectively). **h, i** Representative BSM contraction force traces in response to EFS show that reduced contraction force measured in SMCav1.2$^{+/-}$ BSM tissue, was enhanced by Bay k8644. **j–m** summarized data showing SMCav1.2$^{+/-}$ mice BSM strips exhibited decreased contraction force in response to EFS stimulation (**j** n = 9 BSM strips for both wildtype and SMCav1.2$^{+/-}$ groups), in response to 50 mM KCl (**k** n = 9 BSM strips for both groups), in response to 10 μM carbachol (**l** n = 11 BSM strips for both groups), and in response to 10 μM α,β-meATP (**m** n = 10 BSM strips for both groups). In each condition, contraction was significantly enhanced by 200 nM Bay k8644. Data are shown as box and whiskers, center line is the median of the data set, box represents 75% of the data, and bars indicates whiskers from minimum to maximum. Data are analysed by two tailed Student's t-test. P < 0.05 is considered to be significant and P values are given above the bars. Source data are provided as a Source Data file.

consistently had lower body and bladder weights, resembling ketamine cystitis patients and animal models. TISMCav1.2$^{-/-}$ mouse bladders also show reduced muscle mass and significantly smaller BSM cell size. These bladders exhibited dramatically reduced contractile force in response to EFS, KCl, and agonists carbachol and α,β-meATP, consistent with their diminished Ca$^{2+}$ influx in response to stimulation by Bay k8644 or by KCl (Supplementary Fig. 18). The data conclusively indicate a critical role of Cav1.2 in BSM physiology and recapitulate aspects of the human ketamine cystitis phenotype.

**Cav1.2 agonist Bay k8644 rescues ketamine-induced pathology.** Although Cav1.2 blockers such as nifedipine have been widely used to treat cardiac ischemia and hypertension, trials of Cav1.2 inhibitors have been unsuccessful in treatment of urinary dysfunction[38–41]. In remarkable contrast, our findings demonstrate that Cav1.2 inhibition by ketamine or by nifedipine causes bladder dysfunction. Moreover, Cav1.2 agonist Bay k8644 reverses these in vitro effects of ketamine and nifedipine. We thus hypothesized that Bay k8644 might be used to treat ketamine-induced bladder dysfunction in vivo. As shown in

Fig. 10 and Supplementary Figs. 19, 20, intravesical infusion of ketamine or of nifedipine during CMG significantly increased voiding frequency and decreased peak bladder pressure. Moreover, ketamine and nifedipine each occasionally produced ketamine cystitis-like "bladder spasm" (defined as continuously elevated basal pressure, increased contraction frequency and decreased peak pressure, as demonstrated in Supplementary Fig. 1). All these abnormalities were rapidly abrogated upon instillation of Bay k8644 into the bladder lumen (Fig. 10, in the absence or presence of Cav1.2 antagonists). As described above, heterozygous inactivation of Cav1.2 in smooth muscle also produced a ketamine cystitis-like bladder phenotype, and intravesical infusion of Bay k8644 during CMG corrected the associated voiding abnormalities (Fig. 9d). Intraperitoneal injection of Bay k8644 also corrected these voiding abnormalities (Fig. 9c), including decreased voiding frequency and increased voiding spot size (Fig. 9e–g). In summary, these data confirm Cav1.2 dysregulation as a major mediator of ketamine cystitis, and also confirm Cav1.2 agonist Bay k8644 as a highly effective treatment for experimental ketamine cystitis and potentially other ketamine-associated smooth muscle pathologies.

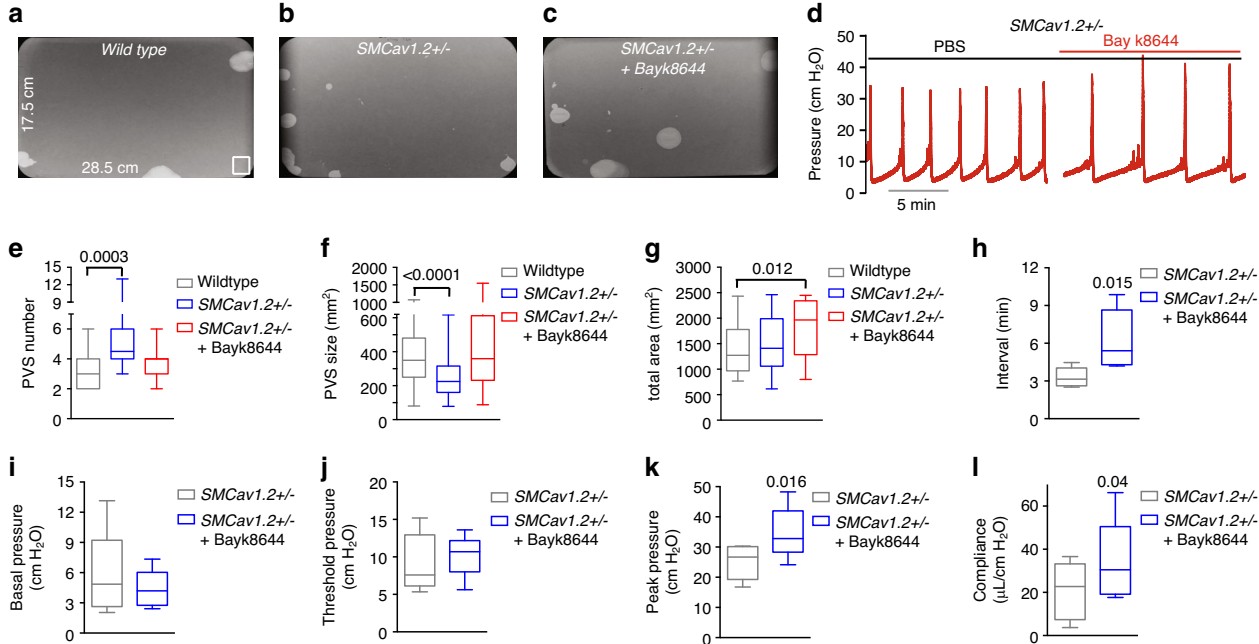

**Fig. 9 SMCav1.2+/− mice exhibit altered urodynamics mimicking ketamine cystitis. a–c** Representative VSA images from *SMCav1.2+/−* mice showing increased number of voids and smaller void size, both corrected by Bay k8644 (2 mg/kg i.p.). **e–g** (*n* = 31 filters for wild type, and *n* = 18 filters for *SMCav1.2+/−* with or without Bay k8644), summarized data indicating *SMCav1.2+/−* mice have increased numbers of primary voiding spots (PVS: voiding spot area ≥80 mm²) and reduced PVS size (**f**). **d** CMG traces from *SMCav1.2+/−* mice. **h–i** (*n* = 5 mice) summarized voiding interval (**h**), basal pressure (**i**), threshold pressure (**j**), peak pressure (**k**), and compliance (**l**), which were quantitated by paired *t*-test. *SMCav1.2+/−* mice showed reduced voiding interval, smaller voiding size, reduced peak pressure and compliance, each rescued by infusion of Bay k8644 (200 nM). Data are shown as box and whiskers, center line is the median of the data set, box represents 75% of the data, and bars indicates whiskers from minimum to maximum. Data are analysed by two tailed Student's *t*-test (**e–g**) and two tailed paired *t*-test (**h–l**). *P* < 0.05 is considered to be significant and *P* values are given above the bars. Source data are provided as a Source Data file.

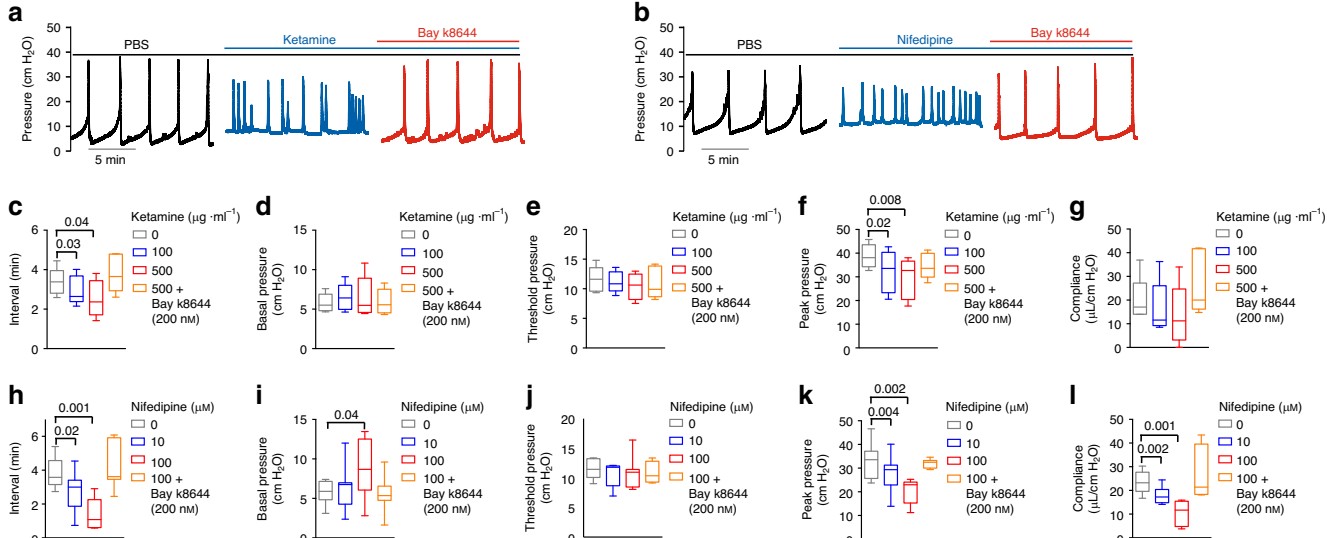

**Fig. 10 Ketamine and nifedipine regulated mouse urodynamics resembling ketamine cystitis in humans. a**, **b** Representative CMG traces of ketamine cystitis-like voiding phenotypes induced by intravesical ketamine (**a** 100, 500 μg ml⁻¹) and nifedipine (**b** 10, 100 μM) are reversed by Bay k8644 infusion (200 nM). Summarized CMG data (**c–l** *n* = 5 mice for ketamine group and *n* = 7 mice for nifedipine group) show that decreased voiding interval (**c**, **h**), peak pressure (**f**, **k**), and compliance (**g**, **l**), were each rescued by Bay k8644 infusion. Data are shown as box and whiskers, center line is the median of the data set, box represents 75% of the data, and bars indicates whiskers from minimum to maximum. Data were analysed by paired two tail Student's *t*-tests. *P* < 0.05 is considered to be significant and *P* values are given above the bars. Source data are provided as a Source Data file.

## Discussion

We have described the Cav1.2 antagonist action of ketamine, and further demonstrated that ketamine inhibition of Cav1.2-mediated signaling plays a major role in the pathogenesis of ketamine cystitis. Ketamine inhibited Cav1.2 expressed in *Xenopus* oocytes, and similarly inhibited L-type $Ca^{2+}$ channel activity in freshly isolated mouse bladder smooth muscle cells. Indeed, ketamine was previously reported to inhibit L-type $Ca^{2+}$ channel activity in bullfrog atrial myocytes[42], guinea pig ventricular myocytes[43], in smooth myocytes of pig tracchea[44], rabbit portal vein[45], and to inhibit KCl depolarization induced $[Ca^{2+}]_i$ elevation in PC12 cells[46]. The broad tissue distribution of Cav1.2 is consistent with the broad spectrum of ketamine's physiological and pathological effects, suggesting that Cav1.2 may serve as a therapeutic target not only in ketamine cystitis, but also for treatment of other smooth muscle-related adverse effects of ketamine.

The antihypertensive properties of L-type calcium channel blockers (CCB) such as nifedipine reflect their inhibition of vascular smooth muscle contraction. As we have shown, CCBs also inhibit contraction of BSM. Indeed, nifedipine was tested as a treatment for overactive bladder patients >40 years ago. These and subsequent tests of additional CCBs to treat bladder hyperactivity or urinary incontinence revealed similar lack of efficacy. Here, we show that although CCB inhibit the magnitude of BSM contractile force, they also promote bladder over-activity and urinary symptoms. This counter-intuitive idea opposes current dogma in clinical urology. However, nifedipine instillation into mouse bladder during CMG not only reduced peak pressure (contractile force), but also caused voiding frequency and even bladder spasm (Fig. 10 and Supplementary Fig. 1), with no instances of increased voiding interval or increased voiding volume at any concentration tested. Indeed, hypertensive patients treated with nifedipine reported increased nocturia averaging four times/night, which ceased or improved after nifedipine was discontinued[47,48]. More recent large-scale clinical studies revealed increased nocturia and urinary urgency and intermittency in CCB-treated patients[49–51]. Consistent with these results, we demonstrated that Cav1.2 agonist Bay k8644 not only reversed the in vitro effects of ketamine and of nifedipine on BSM, but also rapidly and completely reversed ketamine-induced and nifedipine-induced voiding dysfunction in intact mice. Our findings provide mechanistic understanding for the problem of ketamine cystitis, and suggest an approach to clinical therapeutics for ketamine-induced pathologies. These findings also introduce the idea that Cav1.2 agonists of appropriate tissue specificity, rather than (as previously believed) Cav1.2 antagonists such as nifedipine, might effectively treat lower urinary tract syndromes (LUTS). The tissue specificity of alternative splicing of Cav1.2 subunit mRNAs in smooth muscle and cardiac myocytes suggests potential for development of smooth muscle-specific Cav1.2 agonists, perhaps even those selective for bladder smooth muscle over other smooth muscle types[52].

As Cav1.2 plays crucial roles in behavior and nociception, it is also possible that Cav1.2 modulation contributes to the anti-nociceptive and antidepressant effects of ketamine. Recent studies revealed that mice haploinsufficient for Cav1.2 in prefrontal cortex exhibited antidepressant-like behavior[53,54], while human patients with Cav1.2 gain-of-function mutants developed depression-related symptoms[55,56]. Indeed, CCBs such as nifedipine can have antidepressant effects[57–59], whereas treatment of mice with Cav1.2 agonist Bay k8644 induced a depressive-like phenotype with self-injury behavior[60,61]. These reports strongly suggest that our identification of ketamine as an antagonist for Cav1.2 might also provide insight into the antidepressant and antinociceptive actions of ketamine.

## Methods

**Chemicals**. Unless otherwise specified, all chemicals were obtained from Sigma (St. Louis, MO, USA) and were of reagent grade or better. Agonists and antagonists, including α, β-methylene adenosine 5′-triphosphate trisodium (α, β-meATP), Bay K 8644, nifedipine and carbachol were all purchased from Tocris (Minneapolis, MI, USA). Ketamine were purchased from Patterson (Saint Paul, MN, USA). Fluo-4 AM and Pluronic F127 were purchased from Invitrogen (Carlsbad, CA, USA).

**Animal care and use**. Mature female *Xenopus laevis* were obtained from the Dept. of Systems Biology, Harvard Medical School. Frogs were maintained in the AAALAC-accredited animal research facility at Beth Israel Deaconess Medical Center, and subjected to partial ovariectomy under hypothermic tricaine anesthesia. Smooth muscle GluN1 (NR1) homozygous knockout mice (SMNR1KO) and smooth muscle Cav1.2 $\alpha_1$ subunit heterozygous knockout mice (SMCav1.2$^{+/-}$) were generated by crossing mice carrying Cre recombinase under control of the smooth muscle transgelin promoter, (SM22α-creKI, Stock 006878; Jackson Laboratory). Tamoxifen-inducible smooth muscle Cav1.2 knockout mice (TISM-Cav1.2$^{-/-}$) were generated by crossing male smooth muscle myosin heavy chain promoter-Cre male mice (SMMHC-CreER$^{T2}$, Jackson laboratory, Maine) with Cav1.2$^{fl/fl}$ female mice. Since no differences were observed among the NR1$^{fl/fl}$, Cav1.2$^{fl/fl}$, and wild type mice, the results from these control mice were pooled. All mice used in this study were on the C57bl/6j background and were 12–16 weeks' old of age. All animal studies were performed in adherence to U.S. National institutes of Health guidelines for animal care and use, and with the approval of the Beth Israel Deaconess Medical Center Institutional Animal Care and Use Committee.

**Voiding spot assay (VSA)**. Individual mice were gently placed in a standard mouse cage with Blicks Cosmos Blotting Paper (Catalog no. 10422-1005) placed in the bottom for 4 h, during which time water was withheld and standard dry mouse chow was available. Mice were then returned to their home cages and the filter paper was recovered. Filters were imaged under ultraviolet light at 365 nm in a UVP Chromato-Vue C-75 system (UVP, Upland, CA) with an onboard Canon digital single lens reflex camera (EOS Rebel T3, 12 megapixels). Overlapping voiding spots were visually examined and manually separated by outlining and copying, then pasting to a nearby empty space using Fiji software (http://fiji.sc/wiki/index.php/Fiji). Images were analyzed by UrineQuant software developed by us in collaboration with the Harvard Imaging and Data Core. The results table containing the area of each individual voiding spot and the total number of spots, was imported into Excel for statistical analysis. A volume:area standard curve on this paper determined that a 1 mm$^2$ voiding spot represents 0.283 μl of urine. Voiding spots with an area ≥80 mm$^2$ were considered primary voiding spots (PVS), based on voiding spot patterns from hundreds of mice[18,62].

**Myograph**. Bladders were pinned on a small Sylgard block, and muscle was dissected free of the mucosal tissue. BSM strips were then cut longitudinally (2–3 mm wide and 5–7 mm long). Full thickness stomach strips of 5 mm width were cut from the fundus region in the direction of the longitudinal muscle. Full thickness jejunum was cut into ~3 mm rings. Muscle strips were mounted in an SI-MB4 tissue bath system (World Precision Instruments, Sarasota, FL, USA). Force sensors were connected to a TBM 4 M transbridge (World Precision Instruments), and the signal amplified by PowerLab (AD Instruments, Colorado Springs, CO, USA) and monitored through Chart software (AD Instruments). BSM strips were gently prestretched to optimize contraction force, then pre-equilibrated for at least 1 h. All experiments were conducted at 37 °C in physiological saline solution (in mM: 120 NaCl, 5.9 KCl, 1.2 MgCl$_2$, 15.5 NaHCO$_3$, 1.2 NaH$_2$PO$_4$, 11.5 Glucose and 2.5 mM CaCl$_2$) with continuous bubbling of 95% O$_2$/5% CO$_2$. Contraction force was sampled at 2000/s using Chart software. BSM tissue was treated with agonists or antagonists, and/or subjected to electrical field stimulation (EFS).

**Electrical field stimulation**. BSM strip EFS was carried out by a Grass S48 field stimulator (Grass Technologies, RI, USA) using standard protocols[63]. EFS stimulation parameters were set as follows: voltage 50 V; duration: 0.05 ms; trains of stimuli: 3 s; frequencies: 1, 2, 5, 10, 20 and 50 Hz. For stomach muscle strips, EFS stimulation parameters were set as follows: voltage 10 V; duration: 5 ms; trains of stimuli: 8 s; frequencies: 1, 5 10, and 20 Hz.

**Cystometrogram (CMG)**. CMG was performed with PBS infusion (25 μl/min)[64]. Mice were anesthetized by a subcutaneous injection of urethane (1.4 g/kg from 250 mg ml$^{-1}$ solution in PBS) 30–60 min before surgery. At time of surgery, the mouse was further anesthetized with continuous flow isoflurane (3% induction, 1.0% maintenance). Once the pedal reflex was absent, a 1-cm midline abdominal incision was performed. Flame-flanged polyethylene-50 tubing sheathing a 25Gx1.5 in. needle was implanted through the dome of the bladder, then sutured in place (8-0 silk purse string). The incision site was sutured around the tubing using sterile 5-0 silk, and mice were then placed into a Bollman mouse restrainer for 30–60 min stabilization. The catheter was connected to a pressure transducer (and syringe pump by side arm) coupled to data-acquisition devices (WPI Transbridge and AD Instruments Powerlab 4/35) and a computerized recording system (LabChart

software). Bladder filling then commenced, after which voiding occurred naturally through the urethra. Repeated voiding cycles were assessed for change of voiding interval (time between peak pressures), basal pressure (minimum pressure after voiding), threshold pressure (pressure immediately before onset of voiding contraction), peak pressure (maximum voiding pressure minus basal pressure) and compliance (volume, in μl required to increase pressure by 1 cm $H_2O$).

**Primary smooth muscle cell isolation and culture**. All procedures were carried out in a laminar flow hood using sterile techniques. Mouse urinary bladder was placed in a Petri dish (6 cm diameter) containing DMEM supplemented with antibiotics. The adjacent connective and fatty tissues were removed. Urinary bladder was opened longitudinally and bladder neck including trigone and urethra were resected. The detrusor smooth muscle was cut into tiny pieces using surgical scissors. Tissue fragments were washed five times with PBS and digested in 1% collagenase (Gibco, Carlsbad, USA) at 37 °C for 30 min on a shaker. After digestion, enzymes were neutralized by addition of an equal volume of DMEM medium containing 10% FBS. Resulting suspensions were filtered through 100 μm nylon cell strainers (BD, USA) and centrifuged at $180 \times g$ for 5 min. The cell pellet was resuspended in culture medium, and cells were cultured at 37 °C in 5% $CO_2$ and 95% humidity. Growth medium was changed every 2–3 days. Human bladder primary smooth muscle cells (Catalog: FC-0043) were purchased from LifeLine Cell Technology (Frederick, MD) and were cultured in VascuLife SMC culture medium (Catalog: LL-0014) with 5% FBS, 10 mM L-glutamine, 30 μg ml$^{-1}$ ascorbic acid, 5 ng ml$^{-1}$ recombinant human EGF and FGF basic at 37 °C in 5% $CO_2$ and 95% humidity. We used primary cultured mouse bladder smooth muscle cells (1st passage) and primary cultured human smooth muscle cells (3rd–4th passage) for $Ca^{2+}$ imaging and for detection of mRNA and protein. The functional contractile phenotype of these primary cultured smooth muscle cells was confirmed before each experiment.

**Calcium imaging**. Primary cultured BSM cells were seeded at $5 \times 10^3$ cells cm$^{-2}$ onto collagen-coated glass coverslips and maintained for 24–48 h. Before imaging, cultures were washed with physiological saline solution, then loaded with 10 μM Fluo-4 (AM) and 0.02% Pluronic acid for 60 min. After washing cultures twice in PSS, coverslips were placed in a perfusion chamber (Warner instruments, supplied by Harvard Apparatus Ltd., Edenbrige, UK) on the stage of a fluorescent microscope (Olympus, Tokyo, Japan). The chamber was perfused with an automated pump at a flow rate of 1.5 ml min$^{-1}$. Cells were excited at 494 nm (Lambda LS light source; Sutter Instruments, Novato, CA, USA), and the fluorescence emission signal proportional to $[Ca^{2+}]_i$ was measured at 506 nm and images were recorded twice per second. Changes in Fluo-4 fluorescence intensity in all cells in the visual field were acquired with MetaFluor Imaging Software (Molecular Devices). Data were presented as fluorescence intensity. Drug application was regulated by a VC-6 channel valve controller (Warner Instruments) with a Peri-Star perfusion system (WPI).

**Cav1.2 expression in *Xenopus* oocytes**. Plasmids encoding mouse Cav1.2 subunits $\alpha_1$ (Catalog no. 26572), rat $\alpha_2\delta_1$ (Catalog no. 26575), and rat $\beta_3$ (Catalog no. 26574) were purchased from Addgene and sub-cloned into transcription plasmid pXT7 for cRNA transcription and oocyte expression. Mature female Xenopus were anesthetized in a 0.3% tricaine solution. Ovary lobes were surgically isolated from anesthetized frogs and cut into ~1 cm diameter fragments in a standard oocyte solution (in mM: 100 NaCl, 2 KCl, 1.8 CaCl$_2$, 1 MgCl$_2$, 5 HEPES, 2.5 pyruvic acid, and 50 μg ml$^{-1}$ gentamicin; pH 7.6). Follicle membranes were removed by shaking of oocyte clusters treated with collagenase (10 mg ml$^{-1}$, Gibco-BRL, Gaithersburg, MD, USA) in a $Ca^{2+}$-free OR2 solution (in mM: 82.5 NaCl, 2.5 KCl, 1 MgCl$_2$, 5 HEPES; pH 7.6) for 30 min. Stage V-VI oocytes were isolated. Each oocyte was then injected with 6 ng of $\alpha_1$ subunit cRNA, 3 ng of $\alpha_2\delta_1$ cRNA, and 2 ng of $\beta_3$ cRNA to achieve a 1:1:1 molar ratio in a volume of 50 nl using a Drummond Nanoject pipette injector (Parkway, PA, USA) attached to a Narishige micromanipulator (Tokyo, Japan) under a stereo microscope.

**Electrophysiological recording in oocytes**. Oocyte Ba$^{2+}$ currents were recorded at room temperature by two-electrode voltage clamp 3–4 days after cRNA injection, using a GeneClamp 500 two-electrode voltage clamp amplifier (Molecular Devices). Glass microelectrodes of resistance ~1.0 MΩ were pulled from capillaries (Warner Instruments, Hamden, CT, USA) using a P97 pipette puller (Sutter Instrument Co.) and filled with 3 M KCl. Prior to recording, oocytes were injected with ~50 nl of 50 mM BAPTA (1, 2-bis (*o*-aminophenoxy) ethane-*N,N,N,N'*-tetraacetic acid, buffered to pH 7.2) to minimize $Ca^{2+}$-activated Cl$^-$ currents native to oocytes. The bath recording solution contained 10 mM Ba(OH)$_2$, 90 mM NaOH, 1 mM KOH, and 5 mM HEPES (pH 7.4 adjusted with methanesulfonic acid). The currents were sampled at 5 kHz, low pass-filtered at 1 kHz (Digidata 1322 A) using pClamp 8 (Molecular Devices, Palo Alto, CA, USA), and analyzed in Clampfit (Molecular Devices).

**Electrophysiological recording in fresh isolated BSM cells**. Mucosa removed bladder smooth muscle tissue was cut into small strips (5–8 mm long by 2–3 mm wide) and incubated in dissecting solution (in mM: 80 Na Glutamate, 55 NaCl, 6 KCl, 10 glucose, 2 MgCl$_2$, 10 HEPES, final pH 7.3) supplemented with 1 mg ml$^{-1}$

bovine serum albumin (BSA), 1 mg ml$^{-1}$ papain (Worthington Biochemical Corp., NJ), and 1 mg ml$^{-1}$ DL-dithiothreitol (DTT) for ~15 min at 37 °C. Tissues were then transferred to dissecting solution containing 1 mg ml$^{-1}$ BSA, 1 mg ml$^{-1}$ collagenase type II (Life Technologies, CA), 0.5 mg ml$^{-1}$ trypsin inhibitor, and 100 μM CaCl$_2$ for 9–15 min incubation at 37 °C. After digestion, tissues were washed three times with dissecting solution containing 1 mg ml$^{-1}$ BSA and gently triturated with a fire-blunted Pasteur pipette to generate single dissociated BSM cells. Drops of BSM cell suspension were placed into the recording chamber, and cells were allowed to adhere to the bottom for ~20 min before experiment.

Inhibition of Cav1.2 by ketamine was tested in freshly isolated mouse BSM cells by the nystatin-permeabilized cell-attached patch technique for whole cell recording. Patch pipettes were pulled (Sutter Flaming-Brown P97 puller) from borosilicate glass capillaries to tip resistance >3 MΩ when filled with pipette solution. Pipette tips were fire-polished on a microforge (Narashige) before use. For permeabilized cell-attached whole cell recording, the bath solution contained (in mM) 120 NaCl, 20 TEACl, 10 BaCl$_2$, 4 KCl, 10 Hepes, pH 7.4. The pipette solution contained (in mM) 140 CsCl$_2$, 20 KCl, 1 MgATP, 1 EGTA, 10 Hepes, pH 7.2, with addition of 100 μg ml$^{-1}$ nystatin freshly prepared from a 5 mg ml$^{-1}$ stock in methanol. Nystatin permeabilization of the patch membrane after attaining an on-cell gigohm seal established whole cell configuration. Whole cell currents were recorded using an Axopatch 200B amplifier with a Digidata 1440A converter (Axon Instruments/Molecular Devices), using the following voltage pulse protocol. From a holding potential of −70 mV, cells were clamped for 200 ms intervals to test potentials ranging from −80 to +50 mV, then returned to holding potential. Currents were low-pass filtered with a cutoff frequency of 1 kHz, digitized at 2.5 kHz, and stored on a computer. Current densities (pA/pF) were obtained for each cell by normalization of whole cell current to cell capacitance to account for differences in cell membrane surface area. For each cell, capacitive currents measured during 10 ms pulses from a holding potential of −80 mV to a test potential of −75 mV were filtered at a low-pass cutoff frequency of 5 kHz. Data acquisition and analysis was with pCLAMP 11.0.3 software (Axon Instruments/Molecular Devices). Voltage-dependent inactivation was measured from peak inward current during a 500 ms step depolarization to +20 mV following a 4 s prepulse at membrane potentials ranging from −80 to +50 mV in 10 mV increments. Relative peak current ($I/I_{max}$) was fit to a Boltzmann equation.

**RT-PCR**. Total RNA was extracted using the RNeasy kit (Qiagen, Valencia, CA). First-strand cDNA was synthesized from 1 μg total RNA using a SuperScript III First-Strand Synthesis System (Invitrogen). RT-PCR for gene detection was performed with Bio-Rad T100 Thermal Cycler. Quantitative RT-PCR was performed using SYBR Green PCR Mix kits and a 7300 real-time PCR system (Applied Biosystems). *SDHA* (Succinate dehydrogenase complex, subunit A) was used as an internal control. Primer details are provided in Supplemental Table 1.

**Western analysis**. Rabbit anti-α1C antibody (1:1000, #AB5156) was from Millipore. Rabbit monoclonal anti-c-fos (1:1000, #2250) and rabbit monoclonal anti-c-jun (1:1000, #9165) antibodies were from Cell Signaling Technology (Danvers, MA, USA), and rabbit anti-NR1 antibody was from Alomone Lab (#AGC-001). Goat antirabbit Ig-HRP (1:5000, #1705046) was from Bio-Rad (Hercules, CA, USA). Bladder samples, oocytes and cells were lysed in RIPA buffer (150 mM NaCl, 50 mM Tris, 1% v/v NP-40, 0.5% deoxycholic acid, and 0.1% w/v SDS, pH 7.4) containing protease inhibitors (Complete Mini; Roche Applied Science, Indianapolis, IN, USA). Proteins were separated by SDS-PAGE and transferred to PVDF membranes. Membranes were incubated with primary antibody at 4 °C for 1 h after overnight block with 5% nonfat milk. β-actin (Sigma) was used as the internal control. Bands were visualized with Amersham ECL reagent (Arlington Heights, IL, USA). Exposed and developed film was scanned, and the image contrast was corrected with Photoshop (Adobe Systems, San Jose, CA, USA). Images were imported into Adobe Illustrator CS4 (Adobe) for generation of figures. The blots were quantified using FIJI software. Full blots are shown in source data.

**Immunofluorescence staining and confocal microscopy**. Excised bladders were fixed in 4% w/v paraformaldehyde dissolved in 100 mM sodium cacodylate (pH 7.4) buffer for 2 h at room temperature. Fixed tissue was cut into small pieces with a razor blade, cryoprotected, frozen, and sectioned. Cells grown on collagen-coated glass coverslips were fixed for 20 min with 4% (w/v) paraformaldehyde. Tissue or cells were incubated with primary antibodies (1:100) for 2 h at room temperature. After washing away unbound primary antibody, sections were incubated with an Alexa 488-conjugated secondary antibody (diluted 1:100), and actin was stained with rhodamine phalloidin (Cytoskeleton, PHDR1). Tissue or cells were mounted with SlowFade Diamond Antifade Reagent containing DPAI (Thermo Fisher Scientific) to stain nuclei. Images acquired by Zeiss LSM-510 confocal microscope were saved as TIFF files, and imported into Adobe Illustrator CS3. Additional antibodies included rat anti-mouse β1 integrin antibody (BD Bioscience, #550531), anti-c-fos (Cell Signaling, #2250), anti-c-jun (Cell Signaling, #9165), and anti-Ki67 (Abcam, #ab15580).

**Statistical analyses**. All data are expressed as means ± SD, or presented as boxes and whiskers (whiskers extending from minimum to maximum values). Data were

analyzed by Student's $t$-test between two groups or by one-way analysis of variance (ANOVA) for comparison among groups using GraphPad Prism 8 software. If possible, paired $t$-test was used. Tukey HSD (independent) or Bonferroni (paired) multiple comparison post-hoc tests were used where necessary, and $P < 0.05$ was considered significant.

**Reporting summary**. Further information on research design is available in the Nature Research Reporting Summary linked to this article.

## Data availability

Nucleotide sequences of mouse Cav1.2 subunits $\alpha_1$ (Catalog no. 26572), rat $\alpha_2\delta_1$ (Catalog no. 26575), and rat $\beta_3$ (Catalog no. 26574) are available at www.addgene.org. The datasets generated and/or analysed during the current study are available from the corresponding author on reasonable request. Source data are provided with this paper.

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

## Acknowledgements
We thank Drs. Warren Hill (Beth Israel Deaconess Medical Center, Boston), Marcelo Carattino (University of Pittsburgh, Pittsburgh), Robert Moldwin (Northwell Health, Hofstra North Shore-LIJ School of Medicine, New York), Retnagowri Rajandram, and Teng Aik Ong (University Malaya Medical Center, Kuala Lumpur, Malaysia), and Chi Fai Ng (The Chinese University of Hong Kong, Hong Kong) for useful discussions. We thank generous support from Beth Israel Deaconess Medical Center Department of Medicine for this work. We are deeply grateful for the participation of all subjects contributing to this research.

## Author contributions
H.C. performed VSA, myograph, cell isolation, cell culture-based studies, calcium imaging, molecular cloning and expression, PCR & western blot, IF staining and imaging, and analysed data. D.H.V. planned and performed electrophysiological studies and analysed data. X.X. performed VSA. S.L.A. planned and supervised electrophysiological studies. W.Y. performed VSA, CMG, myograph, IF staining and imaging and analysed data. S.L.A. and M.L.Z. contributed to conception and supervision of experiments, and interpretation of data. W.Y. conceived and supervised the project. W.Y. wrote the manuscript, and all authors critically reviewed the manuscript, discussed idea and results, and contributed to the manuscript.

## Competing interests
The authors declare no competing interests.
