## [Peer Review File · Nature Communications]

Reviewers' Comments:

Reviewer #1:

Remarks to the Author:

Chen and co-authors provide evidence that ketamine inhibits CaV1.2 currents and suggest that this mechanism may explain bladder dysfunction in individuals taking ketamine.

Major Comments

As stated by the authors, ketamine concentrations in urine can reach up to 30 ug/ml. In many figures, the concentrations of ketamine used are not stated, making it impossible to review the data. Concentration-response curves to ketamine are shown in supplemental figures, but it is unusual that the authors do not state EC50 values. For the results to be relevant, ketamine concentrations need to be provided for all data. Importantly, when ketamine is administered i.v. or i.m. does it reach concentrations in the blood or urine that inhibit CaV1.2 and cause the bladder phenotype described?

When placed in cell culture, smooth muscle cells switch from a contractile state to a proliferative phenotype. This is associated with a change in ion channel expression profile and a loss of contractility. It has been standard practice for at least two decades to perform experiments using fresh-isolated smooth muscle cells that are harvested on the day of isolation. Here, the isolated smooth muscle cells from mouse bladder have been placed into cell culture for a period of time that is not indicated. Similarly, human bladder smooth muscle cells of unknown passage were purchased commercially and placed into cell culture. Smooth muscle cells shown in figure 2e are clearly of a proliferative phenotype. These cells were used for patch-clamp electrophysiology, calcium imaging experiments and transcription factor studies. All of the experiments with mouse and human need to be repeated using cells isolated on the same day as experiments.

Data shown in figure 3e are not quantified. Quantification needs to be done in order to make statements.

The authors do not describe the mechanism by which ketamine inhibits CaV1.2 channels. This is essential to the study.

The authors claim in several places that they produced bladder smooth muscle-specific CaV1.2 heterozygous (+/-) mice. Rather, they used a constitutive SM22 α -Cre mouse, which raises several serious concerns. First, Cre expression in this model is not bladder smooth muscle-specific. CaV1.2 expression will be reduced in all smooth muscle cells, including vascular, airway, and GI. Second, the Cre model is not inducible and CaV1.2 expression will be reduced in all smooth muscle cells from fertilization, altering many functions, including development. Third, SM22 α is not smooth muscle cell-specific and will reduce CaV1.2 expression in multiple cell types other than smooth muscle, including cardiac and skeletal muscle (Li et al, Circ Res 1996). The authors need to use an inducible, smooth muscle specific Cre model, such as the Myh11-Cre/ERTs, which is the gold standard for these types of studies.

Minor Comments

The order in which figures are discussed is confusing. For example the first figures cited in the Results are Figure 4g, then Supplemental Figure 7.4-7.7. The figure order needs to be arranged.

Reviewer #2:

Remarks to the Author:

Ketamine has rapid-acting and sustained antidepressant effects in treatment-resistant patients with depression, However, ketamine abusers cause ketamine cystitis. The authors discovered that L-type Ca²⁺ channel(Cav1.2) plays a role in the ketamine-induced cystitis. ALthough the results of this study are interesting, the following minor concerns should be addressed.

Minor concerns:

- 1) Detailed pathological data (in vivo treatment) of ketamine and nifedipine should be included.
- 2) Ketamine has two enantiomers, S-ketamine and R-ketamine. Recently, US FDA approved S-ketamine nasal spray for treatment-resistant depression. Did you examine the effects of two enantiomers in Cav1.2 channel?
- 3) Does Cav2.1 channel inhibitory effect of ketamine play a role in the antidepressant action?

Reviewer #3:

Remarks to the Author:

This is an interesting paper providing some new insights into ketamine effects on the bladder. The paper makes a convincing case that at extremely high doses, ketamine inhibits Cav1.2 and thereby reduces bladder contractility and that it has downstream effects on transcription and cellular proliferation. The authors highlight that ketamine-induced bladder irritation is an understudied, common, and important complication of the abuse of ketamine.

I think that it is extremely important to distinguish between the therapeutic effects of ketamine for depression and pain, the anesthetic effects of ketamine, and the consequences of ketamine abuse. To my knowledge, bladder dysfunction is only associated with the abuse of ketamine. This paper makes a compelling case (although it should be stated more clearly) that the antidepressant dose of ketamine (0.1-0.2 mcg/ml) and the anesthetic dose of ketamine (1-2 mcg/ml) do not produce much effect on contractility and, by implication, do not have prominent effects via cav1.2. This hypothesis is consistent with studies in animals and humans that VGCC antagonists attenuate the behavioral effects of R/S-ketamine:

1: Uchihashi Y, Kuribara H, Tadokoro S. Assessment of the ambulation-increasing effect of ketamine by coadministration with central-acting drugs in mice. *Jpn J Pharmacol.* 1992 Sep;60(1):25-31. PubMed PMID: 1460802.

2: Krupitsky EM, Burakov AM, Romanova TN, Grinenko NI, Grinenko AY, Fletcher J, Petrakis IL, Krystal JH. Attenuation of ketamine effects by nimodipine pretreatment in recovering ethanol dependent men: psychopharmacologic implications of the interaction of NMDA and L-type calcium channel antagonists. *Neuropsychopharmacology.* 2001 Dec;25(6):936-47. PubMed PMID: 11750186.

The highest level of ketamine abuse reported among a group with extraordinarily heavy ketamine abuse was about 400x the therapeutic dose of ketamine (15 g/day or about 80 mcg/ml). It is not clear to me that the higher doses of ketamine studied here are relevant to humans.

1: Xu K, Krystal JH, Ning Y, Chen DC, He H, Wang D, Ke X, Zhang X, Ding Y, Liu Y, Gueorguieva R, Wang Z, Limoncelli D, Pietrzak RH, Petrakis IL, Zhang X, Fan N. Preliminary analysis of positive and negative syndrome scale in ketamine-associated psychosis in comparison with schizophrenia. *J Psychiatr Res.* 2015 Feb;61:64-72. doi: 10.1016/j.jpsychires.2014.12.012. Epub 2014 Dec 24. PubMed PMID: 25560772; PubMed Central PMCID: PMC4445679.

This paper leaves a number of issues unresolved: 1) it describes changes consistent with erosion, but it does not define the relationship between these biomarkers and actual mucosal damage, 2) it

only tests racemic ketamine and does not study the relative contribution of R-ketamine and S-ketamine at doses that produce erosions...would use of a single isomer reduce risk? 3) related to point 1, what is the relationship between the biomarkers identified in this study and lymphocytic infiltration of the bladder? and 4) what is the duration of treatment/timecourse that relates the acute physiologic effects, the downstream effects on gene expression and cellular proliferation and the appearance of persistent bladder injury/dysfunction (is there a safe exposure at high doses?).

Minor notes:

HNK does not activate AMPA receptors. It may act via mGluR2 inhibition to increase glutamate release and the released glutamate stimulates AMPA-R. (L 56). Raising the issue of HNK does beg the question of whether any of the key ketamine metabolites are more potent than the parent compound in blocking Cav1.2.

Other NMDA antagonists (dextomethorphan, for example) are abused and also block VGCCs. Why don't they produce the bladder lesions?

Authors' response to reviewer's comments:

We thank our reviewers for their constructive comments.

The initial version of our manuscript was submitted to *Nature*, from which it was transferred to *Nature Communications*, still in *Nature* format (with a "concise description," four multi-panel figures in the main text, and >30 supplemental figures). We apologize for the confusion our reviewers experienced with this initial format.

Our revised manuscript has been reformatted for *Nature Communications*, with 10 figures in the main text, and have made further revisions based on our reviewers' critiques, as summarized in our point-by-point responses below.

Reviewer #1 (Remarks to the Author):

Chen and co-authors provide evidence that ketamine inhibits CaV1.2 currents and suggest that this mechanism may explain bladder dysfunction in individuals taking ketamine.

Major Comments

As stated by the authors, ketamine concentrations in urine can reach up to 30 ug/ml. In many figures, the concentrations of ketamine used are not stated, making it impossible to review the data. Concentration-response curves to ketamine are shown in supplemental figures, but it is unusual that the authors do not state EC50 values. For the results to be relevant, ketamine concentrations need to be provided for all data. Importantly, when ketamine is administered i.v. or i.m. does it reach concentrations in the blood or urine that inhibit CaV1.2 and cause the bladder phenotype described?

Ketamine doses or concentrations were presented in most original figures or figure legends, although concentrations were indeed absent from a few figures. For example, the author's question "*when ketamine is administered i.v. or i.m. does it reach concentrations in the blood or urine that inhibit CaV1.2 and cause the bladder phenotype described?*", was answered in original Fig. 1b. We also stated in the original Results that "our myographic studies detected inhibition of BSM contraction at sub $\mu\text{g/ml}$ concentrations of ketamine (consistent with plasma ketamine concentrations in both ketamine abusers and patients, with full inhibition at $\sim 500 \mu\text{g/ml}$ (Fig. 1a, b)"- line 109-112. Our results and statement are also consistent with reviewer #3's comment that serum ketamine concentrations are at 0.1-0.2 $\mu\text{g/ml}$ for antidepressant dose, and at 1-2 $\mu\text{g/ml}$ for anesthetic dose, and could reach about 80 $\mu\text{g/ml}$ in ketamine abusers - please see reviewer #3's comments and corresponding references. Plasma ketamine concentrations up to 26 $\mu\text{g/ml}$ have also been reported during ketamine anesthesia^{1,2}. Therefore, the ketamine dose response shown in Figure 1b is

relevant to serum ketamine levels measured in human ketamine users and abusers.

The conclusion that Cav1.2 inhibition by ketamine can mediate ketamine-induced pathology is convincingly confirmed by our data that Cav1.2 agonists fully reverse ketamine-induced voiding dysfunction (revised Figure 10). We have added the following sentence in the revised manuscript to clarify this issue (paragraph 2 in Results): "These data are consistent with plasma ketamine concentrations in both ketamine abusers and patients, with plasma concentrations of 0.1-0.2 $\mu\text{g/ml}$ producing analgesia, 0.05-0.2 $\mu\text{g/ml}$ for drowsiness and perceptual distortions, and 2-3 $\mu\text{g/ml}$ for general anesthesia. Plasma ketamine concentrations as high as 26 $\mu\text{g/ml}$ have been reported in anesthesia".

When placed in cell culture, smooth muscle cells switch from a contractile state to a proliferative phenotype. This is associated with a change in ion channel expression profile and a loss of contractility. It has been standard practice for at least two decades to perform experiments using fresh-isolated smooth muscle cells that are harvested on the day of isolation. Here, the isolated smooth muscle cells from mouse bladder have been placed into cell culture for a period of time that is not indicated. Similarly, human bladder smooth muscle cells of unknown passage were purchased commercially and placed into cell culture. Smooth muscle cells shown in figure 2e are clearly of a proliferative phenotype. These cells were used for patch-clamp electrophysiology, calcium imaging experiments and transcription factor studies. All of the experiments with mouse and human need to be repeated using cells isolated on the same day as experiments.

We agree with the reviewer that freshly-isolated smooth muscle cells are often used for smooth myocyte patch clamp studies. However, the reviewer's comment is misdirected insofar as we did not perform patch clamp studies on smooth muscle cells in this manuscript. As described in Methods and Results, we performed voltage clamp studies in *Xenopus* oocytes expressing Cav1.2 from microinjected cRNA.

We also agree with the reviewer's widely accepted observation that during cell culture, smooth muscle cells exhibit a time-dependent shift from a contractile state to a predominantly synthetic phenotype. However, our cultured cells retained their smooth myocyte phenotype in that they maintained spontaneous contractility, and cells prepared and cultured by the same methods have been extensively used by others³⁻⁵. We used primary cultured mouse bladder smooth muscle cells (1st passage) and primary cultured human smooth muscle cells (3rd-4th passage) for our calcium imaging and transcription studies. These cells exhibit contractile/calcium influx responses to carbachol, ATP, and Cav1.2 agonist Bay k8644, as well as inhibitory responses to Cav1.2 antagonist, nifedipine, indicating preservation of basic functional phenotypes of bladder smooth muscle cells. The reviewer also criticized the smooth muscle cells in original Figure 2e as "clearly of

proliferative phenotype" (see revised Fig. 5). This criticism is unjustified because calcium signaling is more prominent in the main cell body compared to branches and peripheral cell area. In revised Figure 7 our rhodamin-phalloidin stained smooth muscle cells demonstrate a typical smooth muscle phenotype. We thus contend that our cell culture studies are appropriate for this manuscript.

Data shown in figure 3e are not quantified. Quantification needs to be done in order to make statements.

We thank the reviewer for noting this omission. We have quantitated the data from original Figure 3e and present that data in revised Figure 6f.

The authors do not describe the mechanism by which ketamine inhibits CaV1.2 channels. This is essential to the study.

Our manuscript has defined ketamine as a novel antagonist of Cav1.2. Ion channel antagonists can act as (1) competitive or (2) non-competitive antagonists, and as (3) pore blockers or (4) allosteric modulators. Ketamine is considered a non-competitive antagonist of the NMDA receptor. Ketamine binds to a site deep within the channel (electrically defined) with properties of use-dependent open-channel block. Ketamine has also been shown to exhibit allosteric modulatory functions. However, the definitive mechanisms by which ketamine inhibits the NMDA receptor remain unclear.

We have provided some preliminary functional evidence in revised Figure 3 that ketamine inhibition of Cav1.2 may be noncompetitive. Although we agree with the reviewer that additional study is required to understand the molecular mechanism by which ketamine inhibits Cav1.2 function, such experiments extend beyond the scope of the current study, which defines Cav1.2 as a major target of ketamine in ketamine-induced pathologies. Indeed, identification of such a target has been a major goal for scientists in the field for the past decade.

In addition to defining ketamine as a novel antagonist for Cav1.2, our manuscript shows that ketamine abolished smooth muscle contractility by inhibiting Cav1.2-mediated Ca^{2+} influx, thus caused voiding dysfunction. The manuscript also demonstrated ketamine inhibition of CaV1.2-mediated accumulation of downstream transcription factors associated with smooth muscle pathological changes. Of yet greater interest is our strong evidence that Cav1.2 agonist Bay k8644 completely abrogated ketamine-induced voiding dysfunction, a finding of great interest for eventual clinical translation. We believe that these data will significantly impact ketamine-related fields.

The authors claim in several places that they produced bladder smooth muscle-specific CaV1.2 heterozygous (+/-) mice. Rather, they used a constitutive SM22a-Cre mouse, which raises several serious concerns. First, Cre expression in this model is not bladder smooth muscle-specific. CaV1.2 expression will be

reduced in all smooth muscle cells, including vascular, airway, and GI. Second, the Cre model is not inducible and CaV1.2 expression will be reduced in all smooth muscle cells from fertilization, altering many functions, including development. Third, SM22a is not smooth muscle cell-specific and will reduce CaV1.2 expression in multiple cell types other than smooth muscle, including cardiac and skeletal muscle (Li et al, Circ Res 1996). The authors need to use an inducible, smooth muscle specific Cre model, such as the Myh11-Cre/ERTs, which is the gold standard for these types of studies.

Reviewer #1 is incorrect in contending that "the authors claim in several places that they produced bladder smooth muscle-specific CaV1.2 heterozygous (+/-) mice." We nowhere in the manuscript claimed to have generated a "bladder smooth muscle specific Cav1.2 knockout". Our original text instead referred appropriately to "smooth muscle-specific Cav1.2+/- knockout"- line 228-247 (see original text p. 10-11).

Our original manuscript cited the inducible Cre model referred to by the reviewer as refs 24 and 25. Indeed, we have also generated this inducible model and have initiated experiments using it. We chose not to include those data because our manuscript is already quite long and data-rich. We are happy to include pertinent data from the inducible knockout model if the editor and reviewers request it.

Minor Comments

The order in which figures are discussed is confusing. For example to first figures cited in the Results are Figure 4g, then Supplemental Figure 7.4-7.7. The figure order needs to be arranged.

Thank you for the comment. We have re arranged the figures.

Reviewer #2 (Remarks to the Author):

Ketamine has rapid-acting and sustained antidepressant effects in treatment-resistant patients with depression, However, ketamine abusers cause ketamine cystitis. The authors discovered that L-type Ca²⁺ channel (Cav1.2) plays a role in the ketamine-induced cystitis. Although the results of this study are interesting, the following minor concerns should be addressed.

Thank you for the favorable comments.

Minor concerns:

1) Detailed pathological data (in vivo treatment) of ketamine and nifedipine should be included.

We and others have previously reported pathological data on Ketamine cystitis in both animal models and humans⁶⁻⁹.

2) Ketamine has two enantiomers, S-ketamine and R-ketamine. Recently, US FDA approved S-ketamine nasal spray for treatment-resistant depression. Did you examine the effects of two enantiomers in Cav1.2 channel?

Thank you for this important question. We recently ordered the R- and S-ketamine enantiomers, and we look forward to answering this question in the near future. However, we believe that answer will be best presented in a separate publication.

3) Does Cav2.1 channel inhibitory effect of ketamine play a role in the antidepressant action?

We indeed raised this question of CaV1.2's possible involvement in ketamine's anti-depressive effect in the original manuscript, but did not investigate it experimentally at that time. The mechanism of ketamine's anti-depressive effects remains unclear, and our manuscript presents a novel molecular target for this interesting area. As noted in our discussion, we speculate that Cav1.2 modulation might indeed contribute to ketamine's anti-depressive action. Further experiments in collaboration with neuroscientists and psychiatrists will soon address this issue.

Reviewer #3 (Remarks to the Author):

This is an interesting paper providing some new insights into ketamine effects on the bladder. The paper makes a convincing case that at extremely high doses, ketamine inhibits Cav1.2 and thereby reduces bladder contractility and that it has downstream effects on transcription and cellular proliferation. The authors highlight that ketamine-induced bladder irritation is an understudied, common, and important complication of the abuse of ketamine.

We thank reviewer for the very positive comments.

I think that it is extremely important to distinguish between the therapeutic effects of ketamine for depression and pain, the anesthetic effects of ketamine, and the consequences of ketamine abuse. To my knowledge, bladder dysfunction is only associated with the abuse of ketamine. This paper makes a compelling case (although it should be stated more clearly) that the antidepressant dose of ketamine (0.1-0.2 mcg/ml) and the anesthetic dose of ketamine (1-2 mcg/ml) do not produce much effect on contractility and, by implication, do not have prominent effects via cav1.2. This hypothesis is consistent with studies in animals and humans that VGCC antagonists attenuate the behavioral effects of

R/S-ketamine:

1: Uchihashi Y, Kuribara H, Tadokoro S. Assessment of the ambulation-increasing effect of ketamine by coadministration with central-acting drugs in mice. *Jpn J Pharmacol.* 1992 Sep;60(1):25-31. PubMed PMID: 1460802.

2: Krupitsky EM, Burakov AM, Romanova TN, Grinenko NI, Grinenko AY, Fletcher J, Petrakis IL, Krystal JH. Attenuation of ketamine effects by nimodipine pretreatment in recovering ethanol dependent men: psychopharmacologic implications of the interaction of NMDA and L-type calcium channel antagonists. *Neuropsychopharmacology.* 2001 Dec;25(6):936-47. PubMed PMID: 11750186.

The highest level of ketamine abuse reported among a group with extraordinarily heavy ketamine abuse was about 400x the therapeutic dose of ketamine (15 g/day or about 80 mcg/ml). It is not clear to me that the higher doses of ketamine studied here are relevant to humans.

1: Xu K, Krystal JH, Ning Y, Chen DC, He H, Wang D, Ke X, Zhang X, Ding Y, Liu Y, Gueorguieva R, Wang Z, Limoncelli D, Pietrzak RH, Petrakis IL, Zhang X, Fan N. Preliminary analysis of positive and negative syndrome scale in ketamine-associated psychosis in comparison with schizophrenia. *J Psychiatr Res.* 2015 Feb;61:64-72. doi: 10.1016/j.jpsychires.2014.12.012. Epub 2014 Dec 24. PubMed PMID: 25560772; PubMed Central PMCID: PMC4445679.

We thank the reviewer for the extensive review of ketamine concentrations and doses associated with therapeutic use and abuse of ketamine in multiple contexts. Please see our response to reviewer #1 for further comments on the issue of dose and concentration. As reviewer #3 notes, this is an extremely important issue related to ketamine use in clinic settings. We agree with the reviewer's assessment of our data suggesting that ketamine dosages used in clinical settings can be "safe". However, side effects, including cardiovascular system and many smooth muscle disorders like nausea, vomiting, and also urinary symptoms, have been reported in clinical ketamine use for analgesic and anti-depression^{8,10-16}. Thus, a single intravenous administration of ketamine (0.75-1.59mg/kg) resulted in urine ketamine concentrations of ~1.5 µg/ml detectable up to 7 days, whereas in some subjects, urine ketamine concentration was below the threshold of detection¹⁷. Another report found urine ketamine concentration in a ketamine abuser can exceed 3µg/ml following ketamine intake of 40mg¹⁸. Clinical dosing of ketamine for pain management can reach up to 1.5g/day¹². This intake, although common among chronic ketamine abusers, which can reach up to 28 g/day¹⁹, with urine ketamine/norketamine concentrations up to 25/50 µg/ml²⁰. Still another clinical study found that 2.2mg/kg intravenous ketamine resulted in plasma ketamine

concentrations up to $\sim 26 \mu\text{g/ml}^1$. Ketamine has also been reported to be hepatotoxic^{12,21-23}. Although most reports of ketamine cystitis note clinical presentation (as opposed to onset of symptoms) between 2-24 months following onset of ketamine abuse, individual cases of cystitis have been observed after a single ketamine exposure.

We believe that Cav1.2 plays an important role in ketamine cystitis and other smooth muscle-related pathologies of ketamine, through both acute inhibition of smooth muscle contractility and chronic modulation of CaV1.2-regulated transcription factors.

This paper leaves a number of issues unresolved: 1) it describes changes consistent with erosion, but it does not define the relationship between these biomarkers and actual mucosal damage, 2) it only tests racemic ketamine and does not study the relative contribution of R-ketamine and S-ketamine at doses that produce erosions...would use of a single isomer reduce risk? 3) related to point 1, what is the relationship between the biomarkers identified in this study and lymphocytic infiltration of the bladder? and 4) what is the duration of treatment/timecourse that relates the acute physiologic effects, the downstream effects on gene expression and cellular proliferation and the appearance of persistent bladder injury/dysfunction (is there a safe exposure at high doses?).

We thank the reviewer for these important questions. (1) Mucosal damage is reported in many ketamine cystitis patients and animal models, but the mechanism is not fully understood. We and others have shown that ketamine-induced voiding dysfunction does not necessarily involve mucosal damage^{6,9,24}, which may be secondary to vascular smooth muscle pathology, possibly induced by contractile overactivity, or other mechanisms which need further study. However, Cav1.2 may be expressed also in urothelium. In addition to ketamine cystitis, other ketamine-induced pathologies also have been reported over the last decade (2) Concerning ketamine enantiomers, please see our response to reviewer #2. (3) Unfortunately, we are currently unable to provide an answer to this question, but hope to find answers in the near future. (4) Please refer to our answer to your dosage question above.

Minor notes:

HNK does not activate AMPA receptors. It may act via mGluR2 inhibition to increase glutamate release and the released glutamate stimulates AMPA-R. (L 56). Raising the issue of HNK does beg the question of whether any of the key ketamine metabolites are more potent than the parent compound in blocking Cav1.2.

We are aware of the possibility that ketamine metabolites might also affect Cav1.2. However, as many of the studies described in our manuscript are

acute in vitro studies with excised bladder or bladder smooth muscle, ketamine metabolism likely contributes little to the measured effects (in this experimental system physically separated from the major sites of metabolism, including liver, kidney, and gut²⁵). Thus ketamine itself likely acts on CaV1.2, but ketamine metabolites should definitely be investigated in this respect. Our recent preliminary myography studies with norketamine and HNK indicate that norketamine but not HNK can also inhibit smooth muscle contraction, suggesting a potential norketamine effect on Cav1.2. If supported by further ongoing studies, the result would suggest the utility of monitoring both ketamine and norketamine in serum and urine in the setting of ketamine administration.

Other NMDA antagonists (dextromethorphan, for example) are abused and also block VGCCs. Why don't they produce the bladder lesions?

Thank you for this interesting question. Bladder pain is indeed listed as a side effect of dextromethorphan (<https://www.drugs.com/sfx/dextromethorphan-quinidine-side-effects.html>). Dextromethorphan has been reported to inhibit mouse and rat bladder smooth muscle contraction²⁶ and to inhibit laboratory animal micturition²⁷. Thus, dextromethorphan may well act on human bladder, but the prevalence and severity of lower urinary tract side effects will be functions of dose, systemic catabolism, and concentrations in urine. The prevalence of potential dextromethorphan lower urinary tract side effects is likely underestimated, if ketamine cystitis is any guide. Since first reported in 2007, epidemiological studies have revealed a ~30% incidence of ketamine cystitis in drug abusers, but only a small fraction of even the severely affected have sought clinical help in clinics.

References

1. Domino, E.F., Zsigmond, E.K., Domino, L.E., Domino, K.E., Kothary, S.P. & Domino, S.E. Plasma levels of ketamine and two of its metabolites in surgical patients using a gas chromatographic mass fragmentographic assay. *Anesth Analg* **61**, 87-92 (1982).
2. Malinovsky, J.M., Servin, F., Cozian, A., Lepage, J.Y. & Pinaud, M. Ketamine and norketamine plasma concentrations after i.v., nasal and rectal administration in children. *Br J Anaesth* **77**, 203-207 (1996).
3. Patel, J.J., Srivastava, S. & Siow, R.C. Isolation, Culture, and Characterization of Vascular Smooth Muscle Cells. *Methods Mol Biol* **1430**, 91-105 (2016).
4. Proudfoot, D. & Shanahan, C. Human vascular smooth muscle cell culture. *Methods Mol Biol* **806**, 251-263 (2012).
5. Zheng, Y., Chang, S., Boopathi, E., Burkett, S., John, M., Malkowicz, S.B. & Chacko, S. Generation of a human urinary bladder smooth muscle cell line. *In Vitro Cell Dev Biol Anim* **48**, 84-96 (2012).
6. Lin, H.C., Lee, H.S., Chiueh, T.S., Lin, Y.C., Lin, H.A., Lin, Y.C., Cha, T.L. & Meng, E. Histopathological assessment of inflammation and expression of

- inflammatory markers in patients with ketamine-induced cystitis. *Mol Med Rep* **11**, 2421-2428 (2015).
7. Liu, K.M., Chuang, S.M., Long, C.Y., Lee, Y.L., Wang, C.C., Lu, M.C., Lin, R.J., Lu, J.H., Jang, M.Y., Wu, W.J., Ho, W.T. & Juan, Y.S. Ketamine-induced ulcerative cystitis and bladder apoptosis involve oxidative stress mediated by mitochondria and the endoplasmic reticulum. *Am J Physiol Renal Physiol* **309**, F318-331 (2015).
 8. Sassano-Higgins, S., Baron, D., Juarez, G., Esmaili, N. & Gold, M. A Review of Ketamine Abuse and Diversion. *Depress Anxiety* **33**, 718-727 (2016).
 9. Rajandram, R., Ong, T.A., Razack, A.H., MacIver, B., Zeidel, M. & Yu, W. Intact urothelial barrier function in a mouse model of ketamine-induced voiding dysfunction. *Am J Physiol Renal Physiol* **310**, F885-894 (2016).
 10. Niesters, M., Martini, C. & Dahan, A. Ketamine for chronic pain: risks and benefits. *Br J Clin Pharmacol* **77**, 357-367 (2014).
 11. Gao, M.R., D.; Liu, H. Ketamine use in current clinical practice. *Acta Pharmacologica Sinica* **37**, 865-872 (2016).
 12. Hocking, G. & Cousins, M.J. Ketamine in chronic pain management: an evidence-based review. *Anesth Analg* **97**, 1730-1739 (2003).
 13. Morgan, C.V., V. Ketamine use: a review. *Addiction* **107**, 27-38 (2011).
 14. Kalsi, S.S., Wood, D.M. & Dargan, P.I. The epidemiology and patterns of acute and chronic toxicity associated with recreational ketamine use. *Emerg Health Threats J* **4**, 7107 (2011).
 15. Gregoire, M.C., MacLellan, D.L. & Finley, G.A. A pediatric case of ketamine-associated cystitis (Letter-to-the-Editor RE: Shahani R, Streutker C, Dickson B, et al: Ketamine-associated ulcerative cystitis: a new clinical entity. *Urology* 69: 810-812, 2007). *Urology* **71**, 1232-1233 (2008).
 16. Storr, T.M.Q., R. Can ketamine prescribed for pain cause damage to the urinary tract? *Palliat Med.* **23**, 670-672 (2009).
 17. Adamowicz, P. & Kala, M. Urinary excretion rates of ketamine and norketamine following therapeutic ketamine administration: method and detection window considerations. *J Anal Toxicol* **29**, 376-382 (2005).
 18. Cheng, W.C., Ng, K.M., Chan, K.K., Mok, V.K. & Cheung, B.K. Roadside detection of impairment under the influence of ketamine--evaluation of ketamine impairment symptoms with reference to its concentration in oral fluid and urine. *Forensic Sci Int* **170**, 51-58 (2007).
 19. Xu, K., Krystal, J.H., Ning, Y., Chen, D.C., He, H., Wang, D., Ke, X., Zhang, X., Ding, Y., Liu, Y., Gueorguieva, R., Wang, Z., Limoncelli, D., Pietrzak, R.H., Petrakis, I.L., Zhang, X. & Fan, N. Preliminary analysis of positive and negative syndrome scale in ketamine-associated psychosis in comparison with schizophrenia. *J Psychiatr Res* **61**, 64-72 (2015).
 20. Chang, T., Lin, C.C., Lin, A.T., Fan, Y.H. & Chen, K.K. Ketamine-Induced Uropathy: A New Clinical Entity Causing Lower Urinary Tract Symptoms. *Low Urin Tract Symptoms* **4**, 19-24 (2012).
 21. Wong, S.W., Lee, K.F., Wong, J., Ng, W.W., Cheung, Y.S. & Lai, P.B. Dilated common bile ducts mimicking choledochal cysts in ketamine abusers. *Hong Kong Med J* **15**, 53-56 (2009).

22. Cheung, T.T., Poon, R.T., Chan, A.C. & Lo, C.M. Education and Imaging. Hepatobiliary and pancreatic: cholangiopathy in ketamine user--an emerging new condition. *J Gastroenterol Hepatol* **29**, 1663 (2014).
23. Lo, R.S., Krishnamoorthy, R., Freeman, J.G. & Austin, A.S. Cholestasis and biliary dilatation associated with chronic ketamine abuse: a case series. *Singapore Med J* **52**, e52-55 (2011).
24. Tan, S., Chan, W.M., Wai, M.S., Hui, L.K., Hui, V.W., James, A.E., Yeung, L.Y. & Yew, D.T. Ketamine effects on the urogenital system--changes in the urinary bladder and sperm motility. *Microsc Res Tech* **74**, 1192-1198 (2011).
25. Edwards, S.R. & Mather, L.E. Tissue uptake of ketamine and norketamine enantiomers in the rat: indirect evidence for extrahepatic metabolic inversion. *Life Sci* **69**, 2051-2066 (2001).
26. Levin, R.M., Whitbeck, C., Sourial, M.W., Tadrous, M. & Millington, W.R. Effects of dextromethorphan on in vitro contractile responses of mouse and rat urinary bladders. *Neurourol Urodyn* **25**, 802-807 (2006).
27. Tillig, B. & Constantinou, C.E. Supraspinal N-methyl-D-aspartate receptor inhibition influences the micturition reflex and function of the upper urinary tract of anesthetized and conscious rats. *Neurourol Urodyn* **22**, 164-175 (2003).

Reviewers' Comments:

Reviewer #1:

Remarks to the Author:

The authors have modified the manuscript in accordance with my previous review. However, there are still some major concerns that need to be addressed.

1) Smooth muscle cell culture radically alters the expression profile of ion channels. For example, Cav1.2 channel expression decreases following cell culture. The author's argument that others have previously used cultured bladder smooth muscle cells does not justify their use here. Many published manuscripts, particularly those that are older, use methods now known to be inappropriate. The use of cultured smooth muscle cells to study contractile regulation by ion channels is one such approach that is now inappropriate. The inclusion of these results reduces the relevance and significance of the work. I disagree with the author's statement that the cells shown in figure 7c "demonstrate a typical smooth muscle cell phenotype". The images shown in figure 7 are actually typical of migratory, proliferative smooth muscle cells. This issue raises major concerns with all of the experiments in this paper done using cultured smooth muscle cells.

2) It is still not clear why the electrophysiology was performed on recombinant Cav1.2 expressed in *Xenopus* oocytes and not on fresh-isolated smooth muscle cells. Cav1.2 channels are composed of multiple different subunits that can exhibit cell-specific expression. The recombinant channels used here that are composed of recombinant mouse alpha1, rat alpha2delta1 and rat beta3 subunits are unlikely to recapitulate those in bladder smooth muscle cells. Cav1.2 currents should have been recorded in fresh-isolated bladder smooth muscle cells for reasons previously stated. These measurements would have then directly related to the other results in the paper.

3) A mechanism for Cav1.2 channel inhibition needs to be provided, as was previously requested. This is essential for a paper proposing to have identified a novel Cav1.2 channel inhibitor. Does ketamine alter Cav1.2 voltage-dependence (activation, inactivation) or is it a pore-blocker? If Bay K8644 abrogates the inhibition, does ketamine bind to the same site as dihydropyridines? These are straightforward experiments to perform.

Reviewer #2:

Remarks to the Author:

All comments have been addressed.

Reviewer #3:

Remarks to the Author:

I appreciate the responses to review.

I still think that the authors could be clearer:

1. That the toxic effects of ketamine on bladder muscle do not occur at antidepressant doses.
2. That the toxicity in question applies only to muscle. In brain, there is evidence that VGCC antagonism can be neuroprotective generally and protective against NMDA-R antagonist neurotoxicity, as in:

Yan, Jia, et al. "Repeated administration of ketamine can induce hippocampal neurodegeneration and long-term cognitive impairment via the ROS/HIF-1 α pathway in developing rats." *Cellular Physiology and Biochemistry* 33.6 (2014): 1715-1732.

Also, I think that there are a number of issues that the authors leave unaddressed that could reasonably be included in this paper, such as R vs. S ketamine.

Authors' response to reviewer's comments:

We thank our reviewers for their constructive comments. We have performed additional experiments and have included new data specified in our point-by-point responses below.

Reviewer #1 (Remarks to the Author):

The authors have modified the manuscript in accordance with my previous review. However, there are still some major concerns that need to be addressed.

1) Smooth muscle cell culture radically alters the expression profile of ion channels. For example, Cav1.2 channel expression decreases following cell culture. The author's argument that others have previously used cultured bladder smooth muscle cells does not justify their use here. Many published manuscripts, particularly those that are older, use methods now known to be inappropriate. The use of cultured smooth muscle cells to study contractile regulation by ion channels is one such approach that is now inappropriate. The inclusion of these results reduces the relevance and significance of the work. I disagree with the author's statement that the cells shown in figure 7c "demonstrate a typical smooth muscle cell phenotype". The images shown in figure 7 are actually typical of migratory, proliferative smooth muscle cells. This issue raises major concerns with all of the experiments in this paper done using cultured smooth muscle cells.

Answer: The reviewer considers primary culture of smooth muscle cells inappropriate for our study because smooth muscle cell Cav1.2 expression levels could decrease during the culture period. We agreed in our previous response that smooth muscle cell phenotype could change during cell culture and passage. Indeed, aspects of freshly isolated smooth muscle cell phenotype change during isolation and culture without passage. We understand that freshly isolated bladder smooth muscle cells are routinely used in the patch clamp studies suggested by the reviewer. However, primary cultured bladder smooth muscle cells are also routinely used for studies of calcium imaging and cell proliferation.

Perhaps more importantly, we are not investigating the number of Cav1.2 channels expressed in bladder smooth muscle cells. Rather, our major question is whether ketamine can inhibit Cav1.2-mediated calcium influx and/or proliferation. Our primary cultured bladder smooth muscle cells exhibit consistently strong contractility in response to Cav1.2 agonist, to the muscarinic agonist carbachol, and to the purinergic agonist ATP, all differentiated characteristics of bladder smooth muscle cells consistent with numerous published studies.

In response to the reviewer's request, we also performed calcium imaging studies on fresh isolated bladder smooth muscle cells. As shown in the following figure, these data in freshly isolated bladder smooth muscle cells fully confirm our earlier finding in primary cultured bladder smooth muscle cells that ketamine inhibits calcium influx stimulated by Bayk8644, by KCl, by carbachol, and by ATP.

Figure, Ketamine inhibits Cav1.2-mediated calcium influx in freshly isolated mouse BSM cells. a - c are representative Fluo-4 Ca²⁺ images of freshly isolated mouse BSM cells treated without or with Bay k8644 (10 nM), or treated first with ketamine (100 μg/ml) and then with added Bay k8644 (10 nM). Ketamine inhibits calcium influx stimulated by Bay k8644 (e), by KCl (f), by carbachol (g), and by ATP (h) in freshly isolated mouse BSM cells.

2) *It is still not clear why the electrophysiology was performed on recombinant Cav1.2 expressed in Xenopus oocytes and not on fresh-isolated smooth muscle cells. Cav1.2 channels are composed of multiple different subunits that can exhibit cell-specific expression. The recombinant channels used here that are composed of recombinant mouse alpha1, rat alpha2delta1 and rat beta3 subunits are unlikely to recapitulate those in bladder smooth muscle cells. Cav1.2 currents should have been recorded in fresh-isolated bladder smooth muscle cells for reasons previously stated. These measurements would have then directly related to the other results in the paper.*

Answer: The reviewer questioned why we use Xenopus oocytes to study ketamine inhibition on Cav1.2 but not freshly isolated smooth muscle cells. We chose to study ketamine inhibition of Cav1.2 in oocytes because they are a well established and widely accepted model for the study of ion channels. Oocytes possess no endogenous background activity, and provide a clear answer to our experimental question. Our choice of Cav1.2 subunits for expression in oocytes is based on subunits expressed in bladder smooth muscle cells. We also cited previous experiments showing that deletion of Cav1.2 causes total loss of bladder smooth muscle contractility.

Nonetheless, in response the reviewer's request, in this R2 revised manuscript we have performed new patch clamp studies on fresh isolated bladder smooth muscle cell. Our data confirm that ketamine inhibits endogenous Cav1.2-mediated currents (Figure 4 f-

h).

3) A mechanism for Cav1.2 channel inhibition needs to be provided, as was previously requested. This is essential for a paper proposing to have identified a novel Cav1.2 channel inhibitor. Does ketamine alter Cav1.2 voltage-dependence (activation, inactivation) or is it a pore-blocker? If Bay K8644 abrogates the inhibition, does ketamine bind to the same site as dihydropyridines? These are straightforward experiments to perform.

Answer: Our new data show that 500 µg/ml ketamine completely inhibits Bayk8644-induced current (Suppl. Fig. 7 a & d), a dosage consistent with ketamine inhibition of Cav1.2 expressed in *Xenopus oocytes* (Fig. 4), inhibition of bladder smooth muscle contraction force (Fig. 1), and on inhibition of Bayk8644-induced bladder smooth muscle contraction (Fig. 3e). Ketamine does not alter voltage-dependence of steady-state currents. Ketamine-mediated inhibition of Cav1.2 currents is due in part to a left-shift in voltage-dependent inactivation without significant change in the time constant(s) of inactivation. These data are included as new Suppl. Figure 7.

Reviewer #2 (Remarks to the Author):

All comments have been addressed.

Thank you very much for your favorable comments.

Reviewer #3 (Remarks to the Author):

I appreciate the responses to review.

I still think that the authors could be clearer:

1. That the toxic effects of ketamine on bladder muscle do not occur at antidepressant doses.

Answer: We provided extensive information in response to this concern in our previous Response to Reviewers and in the R1 manuscript, including: (1) ketamine can be concentrated in the urine. (2) Multiple reports confirm adverse systemic effects of ketamine administration for clinical depression or for chronic pain. (3) ketamine kinetics and metabolism differ significantly from person to person. We are unable to draw this conclusion from this study and make a statement that the toxic effect of ketamine on bladder muscle do not occur at antidepressant doses.

2. That the toxicity in question applies only to muscle. In brain, there is evidence that

VGCC antagonism can be neuroprotective generally and protective against NMDA-R antagonist neurotoxicity, as in:

*Yan, Jia, et al. "Repeated administration of ketamine can induce hippocampal neurodegeneration and long-term cognitive impairment via the ROS/HIF-1 α pathway in developing rats." *Cellular Physiology and Biochemistry* 33.6 (2014): 1715-1732.*

Also, I think that there are a number of issues that the authors leave unaddressed that could reasonably be included in this paper, such as R vs. S ketamine.

Answer: Our current study does not address these points. (1) We have cited references documenting ketamine toxicity has been reported in many organ and tissue systems. (2) We have also cited data showing widespread expression of Cav1.2, including in the central nervous system.

We respectfully submit that the differential effects of R and S ketamine are appropriate subjects of future experiments.

Reviewers' Comments:

Reviewer #1:

Remarks to the Author:

No further comments.

Authors' response to reviewer's comments:

REVIEWERS' COMMENTS:

Reviewer #1 (Remarks to the Author):

No further comments.

We appreciate that the reviewer was positive on our updated data and gave favorable comments.